# Quantifying ice loss in the eastern Himalayas since 1974 using declassified spy satellite imagery

Joshua M. Maurer[1,2], Summer B. Rupper[3], Joerg M. Schaefer[1,2]

[1]Lamont-Doherty Earth Observatory (L-DEO), Palisades, NY 10964, USA
[2]Department of Earth and Environmental Sciences, Columbia University, New York, New York 10027, USA
[3]University of Utah, Department of Geography, Salt Lake City, UT 84112, USA

*Correspondence to*: Joshua M. Maurer (jmaurer@ldeo.columbia.edu)

**Abstract.** Himalayan glaciers are important natural resources and climate indicators for densely populated regions in Asia. Remote sensing methods are vital for evaluating glacier response to changing climate over the vast and rugged Himalayan region; yet many platforms capable of glacier mass balance quantification are somewhat temporally limited considering typical glacier response times. We here rely on declassified spy satellite imagery and ASTER data to quantify surface lowering, ice volume change, and geodetic mass balance during 1974-2006 for glaciers in the eastern Himalayas, centered on the Bhutan-China border. The wide range of glacier types allows for the first mass balance comparison between clean, debris, and lake-terminating (calving) glaciers in the region. Measured glaciers show significant ice loss, with an estimated mean annual geodetic mass balance of $-0.13 \pm 0.06$ m.w.e. $\text{yr}^{-1}$ (meters of water equivalent per year) for 10 clean-ice glaciers, $-0.19 \pm 0.11$ m.w.e. $\text{yr}^{-1}$ for 5 debris-covered glaciers, $-0.28 \pm 0.10$ m.w.e. $\text{yr}^{-1}$ for 6 calving glaciers, and $-0.17 \pm 0.05$ m.w.e. $\text{yr}^{-1}$ for all glaciers combined. Contrasting hypsometries along with melt pond, ice cliff, and englacial conduit mechanisms result in statistically similar mass balance values for both clean-ice and debris-covered glacier groups. Calving glaciers comprise 18% (66 $\text{km}^2$) of the glacierized area, yet have contributed 30% (-0.7 $\text{km}^3$) to the total ice volume loss, highlighting the growing relevance of proglacial lake formation and associated calving for the future ice mass budget of the Himalayas as the number and size of glacial lakes increase.

## 1 Introduction

Glaciers in high mountain Asia hold the largest store of ice outside the Polar Regions and contribute meltwater used by roughly 20 percent of the world's population for agriculture, energy production, and potable water (Immerzeel et al, 2010). Glacier changes must be quantified in order to evaluate impacts to hydrology and ecosystems, assess glacial lake outburst flood (GLOF) hazards, calculate recent contributions to sea level rise, and increase predictive capabilities regarding future change and resulting impacts.

The heterogeneity of Himalayan glaciers poses significant challenges in quantifying and modelling glacier changes. Debris-cover in the ablation zone and calving in proglacial lakes are particularly noteworthy examples of complicating factors that may significantly affect the response of many glaciers. Bolch et al (2012) estimate the proportion of debris-

covered ice in the Himalayas is ~10%, and Scherler et al (2011) estimate that 93% of glaciers in the Himalayas have > 20% debris-covered areas. Debris-covered glaciers are difficult to model, since debris can either increase or suppress melt depending on debris thickness and extent, though debris-covered glaciers in the Himalayas are mostly assumed to be less responsive to ongoing warming (Scherler et al, 2011). Similarly, numerical models of glaciers terminating in moraine-dammed proglacial lakes are poorly constrained, and these glaciers can undergo enhanced ice loss through calving and thermal-undercutting processes independent of climate. Lake-terminating glaciers have particular societal relevance, because the growing lakes can cause GLOFs (Glacial Lake Outburst Floods), as well as impact glacier mass balance and hydrology.

Response of Himalayan glaciers to changing climate remains somewhat controversial, primarily due to this complexity of the glacier systems combined with scarcity of direct observation, and no unambiguous pattern has emerged (Berthier et al, 2007; Kääb et al, 2012). Complex politics, rugged terrain, and the immense number of glaciers result in a severe lack of field data (Bolch et al, 2011; Rupper et al, 2012). The few available field records in the Himalayas are often biased towards small to medium-sized clean-ice glaciers due to logistical reasons, and tend to be located in sub-regions where mass loss is greater than in the region as a whole (Gardner et al, 2013). Thus there is a critical need for spatially and temporally extensive glacier change data that captures the full spread of glacier complexities over timespans relevant to glacier response times. Here we focus on the eastern monsoonal Himalayas, centered on the Bhutan-China border. Few data on glacier changes are available and the region has a large diversity of glaciers with regard to elevation, size, debris-cover, hypsometry, accumulation rates and calving characteristics. Furthermore, glacier meltwater is an important hydrological resource in Bhutan, including for hydroelectric power generation (Beldring and Voksø, 2012). Recent hydrologic mixing model results using isotopic and geochemical chemistry have estimated glacier seasonal outflow contributions to the Chamkar Chhu river in Bhutan ranging from ~76% at 4500 m to 31% at 3100 m elevation during September (post-monsoon) (Williams et al, 2015).

There are several clean glaciers flowing northward onto the Tibetan Plateau with high velocities, likely with large amounts of basal sliding (Kääb, 2005). On the southern flank, most large glaciers are debris-covered, located in steep valleys, and show slow-to-nearly stagnant velocities, with many depressions and melt pond features. Additionally, several glaciers (including clean and debris-covered) terminate in moraine-dammed lakes. The majority of clean-ice glaciers in the Bhutan-centered region have a high mass turnover, with most accumulation and ablation occurring during the summer months as a result of the Indian monsoon (Rupper et al, 2012). In regions with high accumulation, ablation is dominated by melt and thus particularly sensitive to changes in temperature (Rupper and Roe, 2008). Robust melt models indicate these eastern-Himalayan, temperature-sensitive glaciers are currently out of balance with present climatology. One estimate predicts a loss of almost 10% of the current glacierized area, with an associated drop in meltwater flux of as much as 30% even if climate were to remain at the present day mean values (Rupper et al, 2012). Kääb et al. (2012) computed a 2003-2008 specific mass balance of -0.26 ± 0.07 to -0.34 ± 0.08 m.w.e. yr$^{-1}$ (depending on different density scenarios for snow and ice) for eastern Nepal and Bhutan using laser altimetry, while Gardelle et al. (2013) estimated a 1999-2011 geodetic mass

balance of $-0.22 \pm 0.12$ m.w.e. $yr^{-1}$ for Bhutan by differencing SPOT5 and SRTM DEMs (digital elevation models). Another recent study utilizing multi-temporal Landsat images to compute glacier area changes in Bhutan showed $23.3 \pm 0.9\%$ glacial area loss between 1980 and 2010, with loss mostly observed below 5600 m a.s.l., and greater area loss for clean glaciers (Bajracharya et al, 2014). The first decadal mass-balance record of the Gangju La glacier in the Bhutan Himalaya was recently published, in which a mass balance of $-1.12$ to $-2.04$ m.w.e. $yr^{-1}$ was estimated between 2003 and 2014 (Tshering and Fujita, 2015).

To build on these pioneering studies and further constrain glacier changes, we utilize a new pipeline (Maurer and Rupper, 2015) to extract DEMs from declassified Hexagon imagery (1974) and ASTER scenes (2006), then use DEM differencing to measure ice volume change and geodetic mass balance between 1974 and 2006 in this temperature-sensitive, monsoon-influenced region. Our approach provides high spatial detail and longer temporal range compared to previous measurements, and circumvents significant uncertainties regarding SRTM radar penetration in ice and snow (Gardelle et al, 2012).

## 2 Methods

Previous studies have demonstrated the value of declassified spy satellite imagery for glacier mass and volume change quantification in various regions of the Himalayas (Lamsal et al, 2011; Bolch et al, 2011; Pieczonka et al, 2013; Bhambri et al, 2013; Raj et al, 2013; Racoviteanu et al, 2014; Pieczonka and Bolch, 2015; Holzer et al, 2015; Pellicciotti et al, 2015; Ragettli et al, 2016). We rely here on a new workflow called HEXIMAP (Hexagon Imagery Automated Pipeline) which utilizes computer vision algorithms to extract DEMs with high efficiency and accuracy (Maurer and Rupper, 2015). Both Hexagon and ASTER DEMs are extracted using similar methods in order to minimize unwanted elevation bias caused by different image processing techniques. The resulting elevation change maps are used to compute average surface lowering of glaciers, changes in ice volume, and geodetic mass balance.

### 2.1 Hexagon

The Hexagon program consisted of a series of 20 photographic reconnaissance satellite systems developed and launched by the United States, operational from 1971 to 1986 during the Cold War era. Each satellite carried approximately 96.5 km of film, and thousands of photographs worldwide were acquired using the mapping camera system (ground resolution of 9 meters, improved to 6 in later missions). The Hexagon mapping camera system acquired multiple 3400 $km^2$ frames as the satellite proceeded along its orbital trajectory. After image acquisition, film-recovery capsules were ejected from the satellite and parachuted back to earth over the Pacific Ocean, where they were retrieved midair via "air snatch" by C-130 Air Force planes. The images have overlap of approximately 55 to 70%, which allows for extraction of digital elevation models (Oder et al, 2012; Surazakov and Aizen, 2010).

Eight separate 5000 x 5000 pixel blocks are processed, which correspond to approximately 20 x 20 km regions with the Hexagon film scanned at 7 μm resolution (orange outlines in Fig. 2), with blocks selected to maximize coverage of glaciers and avoid regions with cloud cover. The same HEXIMAP (Hexagon Imagery Automated Pipeline) methodology, outlined in Maurer and Rupper (2015) is used to extract Hexagon DEMs. In summary, epipolar images are generated by computing homography transformations via automated point detectors and descriptors (Bay et al, 2006; Fusiello and Irsara, 2008). After epipolar resampling, image features line up horizontally and the disparity search is reduced to one horizontal dimension. Disparity maps are computed using the semi-global block matching algorithm (Hirschmüller, 2008), bundle adjustments are performed to minimize reprojection error, and stereo-matched points are triangulated in 3D space using the direct linear method (Hartley and Zisserman, 2003). The 3D point clouds are registered to the reference SRTM DEM using nonlinear optimization of pose parameters including rotation, translation, and a global scale factor. ICIMOD glacier outlines are utilized to exclude glacial terrain during optimization (after being manually edited to match the glacier extent in 1974). The glacier outlines are first converted to a raster binary mask to match the spatial resolution of the reference DEM. Next, dilation (a morphological operation which adds pixels to edge boundaries) is performed to slightly enlarge the glacier boundaries in the raster mask, which helps to eliminate unstable glacier pixels not contained by the glacier outlines, as well as eliminate possibly unstable moraines (see Fig S6). Any elevation change pixels outside of 3 standard deviations are excluded during each iteration in the optimization routine, which effectively eliminates other unknown sources of error during optimization. The higher spatial-resolution Hexagon point clouds are then resampled to match ASTER data (~30 m intervals) using linear interpolation. Data voids larger than 2 km$^2$ are not interpolated to avoid biasing glacier change results with unrealistic terrain (Fig. S3-S5).

**2.2 ASTER**

The Advanced Spaceborne Thermal Emission and Reflection Radiometer (ASTER) was launched on board NASA's Terra spacecraft in December, 1999 as part of a cooperative effort between NASA and Japan's Ministry of Economic Trade Industry (METI). In the visible and near-infrared (VNIR) spectral region (0.78-0.86 μm), ASTER has a nadir view telescope as well as a backward looking telescope to provide stereoscopic capability at 15m ground resolution. Both use 4000 element charge-coupled detectors (CCD's), acquiring data via linear pushbroom scanning. Each ASTER scene covers approximately 60 x 60 km (Abrams, 2000). Although a global ASTER DEM (GDEM v2) is publicly available, anomalies and artifacts in the data limit its utility for glacier change quantification. Instead, two ASTER Level-1A scenes (granule ID: ASTL1A 0612040446230612070303 and ASTL1A 0602030445410602060303) were downloaded from the GDS (Ground Data Systems) ASTER/PALSAR Unified Search website, maintained by Japan Space Systems. DEMs were extracted from the scenes using similar methodology as previously described for the Hexagon imagery, with a few key differences. First, DN (digital number) pixel values from the VNIR images are converted to radiance and processed to remove residual striping artifacts. Second, since ASTER images are acquired by a linear pushbroom sensor they do not have a single fixed center of perspective (Kim, 2000). Consequently, epipolar images cannot be generated using a single homography transformation as

was done with Hexagon images. Alternatively, sight vectors and satellite position matrices (supplied with ASTER ephemeris data) for each CCD row are used to project ASTER forward and backward looking images to a common image plane, after which corresponding pixels in the stereo images are matched using the same stereo-matching algorithm in HEXIMAP. Lastly, point clouds are triangulated by computing sight vector intersections in 3D space rather than using the direct linear method. All other aspects regarding DEM extraction are identical to the Hexagon methodology, thus minimizing any unwanted potential elevation bias caused by different image processing techniques.

## 2.3 Geomorphic Change

To compute glacier changes, the 1974 Hexagon DEMs are subtracted from the 2006 ASTER DEMs to create elevation change maps. Pixels located in areas with > 30° slope are excluded from analysis due to lower accuracies during the DEM extraction process (Maurer and Rupper, 2015). Elevation changes over 100 meters are likely due to stereo matching errors from cloud cover or low radiometric contrast, and are thus also excluded. To delineate glacier boundaries, polygons representing glacier outlines were downloaded from the ICIMOD mountain geoportal (http://geoportal.icimod.org/), which were found to have comparatively greater detail and accuracy for this region compared to the Randolph Glacier Inventory, based on overlay and visual inspection of Google Earth imagery. The polygons were then manually edited to reflect the spatial extent of glaciers in 1974 and 2006 based on visual interpretation of the Hexagon and ASTER imagery, along with examination of the elevation change maps. The 30° slope threshold also effectively excludes any steep parts (nunataks and rock cliffs) in accumulation regions which were erroneously delineated as glacier ice, and the glacier outlines are updated accordingly.

Relative vertical errors between the Hexagon and ASTER DEMs are expected due to different sensor characteristics such as viewing geometry, sun position, cloud cover, and atmospheric conditions. Further complicating this are non-ice geomorphic changes such as landslides, which can be triggered as glaciers recede and alter stress regimes along valley walls and moraine ridges, exposing unstable slopes, and reorganizing large volumes of unconsolidated sediment (Richardson and Reynolds, 2000). Several collapsed moraines were observed in the region (Fig. S6), therefore we base our clean-ice and calving glacier outlines primarily on visible satellite imagery, with the thickness change maps as a secondary source. On the other hand, we base our debris-covered glacier outlines primarily on the thickness change maps, as debris covered glaciers are difficult to distinguish from surrounding terrain using visible imagery only. Future work could focus on utilization of SAR glacier tracking methods to further constrain the extent of debris-covered zones, including surface feature tracking, SAR interferometry, and coherence tracking (e.g. Mattar et al, 1998; Strozzi et al, 2002; Atwood et al, 2010; Frey et al, 2012; Schubert et al, 2013).

Glacier elevation models extracted using stereo photogrammetry often have errors and gaps over snow-covered accumulation zones due to low radiometric contrast and sensor oversaturation (Pellikka and Rees, 2009). Hexagon film strips are especially vulnerable to this problem, resulting in large regions of missing data and some apparently erratic thickness changes over glacier accumulation zones. To exclude these erroneous regions, we compute the neighborhood

standard deviation of each image pixel (a measure of local image contrast, using a 5x5 window), along with the gradient and curvature of the thickness change map for each glacier. Pixels with neighborhood standard deviations less than 3, which also have either a thickness change gradient > 45, or a curvature value > 0.005 m$^{-1}$ are excluded, and gaps in the thickness change maps smaller than 2 km$^2$ are interpolated (Fig. S3-S5). This method allows for removal of erroneous pixels in low-contrast accumulation zones, while retaining pixels in debris-covered zones which often have greater local gradient and curvature values due to melt ponds and ice cliffs.

To close remaining data gaps in accumulation regions, various approaches can be found in the literature. Gardelle et al (2013) replace missing thickness change data over glaciers by the regional mean of the corresponding elevation band for a given glacier type, based on the assumption that thickness changes should be similar at a given altitude across the region. Pieczonka and Bolch (2013) assume no change in the accumulation regions and replace missing data values by zero. Due to the spatially heterogeneous nature of glacier changes in Bhutan, and the limited number of contributing pixels at high elevation bands (Fig. 4), the regional extrapolation method introduces significant bias, especially regarding the large region of missing data in the accumulation zone of glacier c. Thus, all regional glacier change values reported in the text are derived using the method which assumes zero change for missing data. To examine the effect on geodetic mass balance and facilitate a comparison between the two methods, we also include separate results derived using each assumption (replacing missing data with zero change vs. regional extrapolation) in Table S3. For the extrapolation method, missing data for different glacier types are extrapolated using the corresponding thickness change profiles (either clean, debris, or calving).

A total of 21 glaciers are selected (Fig. 2, outlined in white) based on size (glaciers larger than 3 km$^2$), and data coverage (glaciers with at least 25% glacier area covered by the DEMs). Unfortunately, incomplete coverage of remote sensing data, clouds, and poor radiometric contrast in some areas prevent accurate investigation of all glaciers. While this does limit direct comparison to other previous studies which may measure all glaciers in a region, these 21 largest glaciers give a good regional picture of thickness changes over the 3-decade timespan.

Debris-covered areas for each glacier are delineated using a Landsat TM4/TM5 DN band ratio image with a threshold of 2.0 (Paul, 2000). Non-calving glaciers with 20% or greater debris-covered area are assigned the debris category (5 glaciers); non-calving glaciers with less than 20% are assigned the clean category (10 glaciers). The calving category (6 glaciers) includes both clean and debris types which terminate in lakes as determined by viewing the Hexagon and ASTER imagery.

For each glacier, the ice volume change, spatially-averaged thickness change, and geodetic mass balance over the 32-year timespan are computed using the elevation change maps following Eq. (1-3):

$$\Delta V = \sum_{i=1}^{n} D_i r^2, \tag{1}$$

$$\bar{h} = \frac{\Delta V}{A}, \tag{2}$$

$$\dot{b} = \bar{h}\rho, \tag{3}$$

where $\Delta V$ is ice volume change (m³), $D_i$ is the elevation change (m) for pixel $i$ located within a glacier polygon, $n$ is the total number of pixels within a glacier polygon, $r$ is the resolution of the elevation change map (~30 m), $\bar{h}$ is the spatially-averaged elevation change of the glacier, $A$ is the average of the 1974 and 2006 glacier areas (m²), $\dot{b}$ is the geodetic (specific) mass balance, and $\rho$ is the estimated average density of glacier ice; here we use an intermediate value between firn and ice of 850 ± 60 kg/m³ as recommended by Huss (2013). Geodetic mass balance values are converted to m.w.e. (meters water equivalent) by dividing $\dot{b}$ by the density of water (1000 kg/m³).

## 2.4 Relative accuracy between DEMs and glacier change uncertainties

Statistical significance of elevation changes are quantified by estimating the relative vertical accuracy between the Hexagon and ASTER DEMs. Table S1 shows the root-mean-square error, mean, median, normalized median absolute deviation, standard deviation, 68.3% quantile, and 95% quantile of elevation changes between each approximately 20 by 20 km Hexagon DEM (orange outlines in Fig. 2) and the ASTER DEM (blue outline in Fig. 2) for assumed stable (ice-free) terrain. Plots of elevation change against elevation, slope, curvature, ASTER along-track and cross-track were also examined for potential biases (Fig. S1). We neglect any global corrections, as the vast majority of data lie in regions with close to zero bias, and pixels with high slope (> 30°) are excluded as outlined in section 2.3.

To assess uncertainties for glacier changes, the point elevation error ($E_{pt}$) and extrapolation error ($E_{ext}$) are used to calculate the total elevation error ($E_z$) for a given elevation band (Nuth et al, 2010):

$$E_z = \sqrt{\left(\frac{E_{pt}}{\sqrt{n_z}}\right)^2 + \left(\frac{E_{ext}}{\sqrt{n_z}}\right)^2}, \tag{4}$$

The standard deviations of the relative elevation change over stable terrain are used for $E_{pt}$ (Table S1), while the standard deviations of glacial elevation change within each 100 m elevation band are used as approximations for $E_{ext}$. These $E_{ext}$ estimates are conservative because the elevation bands contain intrinsic natural variability, as not all glaciers have undergone the same elevation change at a given elevation (Gardelle et al, 2013). The $n_z$ value represent the number of independent pixel measurements. To account for spatial autocorrelation, we estimate $n_z$ as:

$$n_z = \frac{n_b * r^2}{\pi * d^2}, \tag{5}$$

where $n_b$ is the number of pixels in a given glacier elevation band, $r$ is the pixel resolution (~ 30 m), and $d$ is the distance of spatial autocorrelation (Nuth et al, 2010; Nuth and Kääb, 2011). For glacier regions where data exists (i.e. covered by an elevation change map, thus no extrapolation is necessary), $E_{ext}$ is set to zero and the numerator in Eq. 5 is set to the area within the glacier covered by elevation change data. To estimate $d$, we perform a semivariogram analysis, which relates variance to sampling lag and gives a picture of the spatial dependence of each data point on its neighbour (Curran, 1988; Rolstad et al, 2009). For all eight regions, we find the range varies from approximately 300 to 450 m, and thus choose a conservative value of 500 m for $d$. The volume change error for a given glacier is then estimated as:

$$E_{vol} = \sqrt{\sum_1^z (E_z * A_z)^2}, \tag{6}$$

where $A_z$ is the area of the glacier within a given elevation band $Z$. $E_{vol}$ is then combined with glacier area uncertainties of $\pm$ 10% and an ice density uncertainty of $\pm$ 60 kg/m$^3$ using standard quadratic (uncorrelated) error propagation. All final glacier change uncertainties are reported as $\pm$ 1 SEM (standard error of the mean) unless noted otherwise.

## 3 Results

Probability density plots of regional elevation change between the years 1974-2006 yield a negatively-skewed distribution for glaciers with a mean of -11 m and a standard deviation of 20 m, reflecting the approximate span of ice surface lowering. The surrounding ice-free terrain shows a narrower distribution centered near zero, with a mean of 0.7 m and a standard deviation of 10 m (Fig. 1). Non-zero elevation change values in the ice-free terrain distribution (blue region, Fig. 1) are likely caused by a combination of actual changes such as landslides, errors caused by clouds, and other intrinsic errors

associated with stereo photogrammetric methods used to create the DEMs.

    All clean, debris, and calving glaciers investigated here for change during the 32-year timespan show predominate lowering and retreat of ice surfaces (Figs. 2 and 3). Individual glacier change statistics are also given in Table S2, including ice volume changes, spatially averaged thickness changes, and geodetic mass balances.

    The relatively consistent negative mass balance trend includes both clean and debris-covered glaciers. Further insight

into the ice-loss patterns can be obtained by examining the elevation change maps (Fig. 3). Most clean glaciers are retreating and exhibit thinning near their toes. Conversely, the debris-covered glaciers exhibit irregular patterns of elevation loss in their ablation area. Several smaller debris-covered glaciers have varying amounts and distributions of debris, and show different patterns of thinning. Some glaciers show the greatest thinning near their toes, others exhibit downwasting in mid-section of the glacier, and still others display scattered ice-loss features. Ice loss is greatly enhanced for several glacier toes

terminating in moraine-dammed lakes.

The mean (area-weighted) geodetic mass balance for the selected glaciers (Fig. 2, outlined in blue) is estimated to be -5.4 ± 1.6 m.w.e. for the period 1974 to 2006. Averaged over the 32-year timespan, this yields an annual mass balance of -0.17 ± 0.05 m.w.e. yr$^{-1}$. Clean glaciers comprise 61% (221 ± 11 km$^2$) of the total studied glacierized area (365 ± 12 km$^2$ for 21 glaciers), and have contributed 46% (1.09 ± 0.4 km$^3$) to the total ice volume loss with a mass balance of -0.13 ± 0.06

m.w.e. yr$^{-1}$. The debris glaciers comprise 21% (78 ± 4 km$^2$) of the total glacierized area, and have contributed 24% (-0.55 ± 0.4 km$^3$) to the total ice volume loss with a mass balance of -0.19 ± 0.11 m.w.e. yr$^{-1}$. Calving glaciers comprise 18% (66 ± 3 km$^2$) of the total glacierized area, and have contributed 30% (-0.70 ± 0.3 km$^3$) to the total ice volume loss with a mass balance of -0.28 ± 0.10 m.w.e. yr$^{-1}$.

## 4 Discussion

## 4.1 Regional Glacier Change

The regional mass budget result of -0.17 ± 0.05 m.w.e. yr$^{-1}$ from 1974-2006 is less negative than other estimates derived from remote sensing over shorter time periods. For example, Gardelle et al. (2013) reported a mass budget of -0.22 ± 0.12 m.w.e. yr$^{-1}$ during 1999-2011, and recalculated the Kääb et al (2012) results to obtain -0.52 ± 0.16 m.w.e. yr$^{-1}$ during 2003-2008 for the Bhutan region. Additionally, our estimate is significantly less negative compared to the 1970-2007 mass budget

of 0.32 ± 0.08 m w.e. a$^{-1}$ in the neighbouring Everest region estimated by Bolch (2011). We hypothesize that the shorter more recent timespans of the Kääb et al (2012) and Gardelle et al (2013) studies result in more negative mass budgets due to accelerating glacier retreat in Asia since the end of the 1970's (Zemp et al, 2009). Addition influencing factors include different spatial extents covered, radar penetration uncertainties involved with SRTM data (not an issue in this study), and different methods of dealing with data gaps in accumulation zones.

Table S3 gives results obtained using the two different gap-filling methods in accumulation zones. Both methods yield similar geodetic mass balance values when glacier c (which has a disproportionately large region of missing data at high elevation) is not extrapolated, and purely by chance add up to exactly similar values for ΔV and b˙ in the "all" category. When glacier c is extrapolated using the limited number of contributing pixels at high elevation, it introduces significant unrealistic bias which overshadows the measured ice thickness changes of other glaciers, making the regional mass balance values

unrealistically positive. This illustrates that care must be taken when extrapolating from individual elevation bands from regional profiles, to avoid extrapolating large regions from a few unreliable data points.

Using a degree-day melt model, Rupper et al. (2012) estimated an area-averaged, net mass balance of -1.4 ± 0.6 m.w.e. yr$^{-1}$ (averaged over the time period 1980-2000) for the entire glacierized area (glaciers and perennial snowpack) of the Bhutanese watershed. Recently published in-situ measurements of –1.12 to –2.04 m.w.e. yr$^{-1}$ between 2003 and 2014 for the

Gangju La glacier (located approximately 15 km southwest from the toe of glacier d in Fig. 2) agree well with the melt model results (Tshering and Fujita, 2015). Compared to the remote sensing estimates, the modelled and in situ results are significantly more negative. Though difficult to compare regional changes to local ones, Cogley (2012) suggest that the

discrepancy between in situ vs. remote sensing measurements may be explained by the smaller size and lower elevations of glaciers selected for fieldwork, along with unquantified local factors such as mass gain by snow avalanching. Regarding the melt model, it does not account for the insulating effects of debris-cover, while accounting for the albedo effects of the debris, which would lead to a significant overestimation of modeled melt over debris-covered glaciers. For clean ice glaciers and perennial snow, the modeled net mass balance is considerably less negative, $-0.3 \pm 0.2$ m.w.e. $yr^{-1}$. This latter value is more consistent with our geodetic mass balance of $-0.17 \pm 0.05$ m.w.e. $yr^{-1}$ presented here, yet still on the high end. Taken together, the remote sensing data support a more conservative model scenario of future glacierized area loss and meltwater flux change, highlight the benefit of informing modeling and in-situ approaches with remote sensing, and exemplify the need for further understanding of these discrepancies.

## 4.2 Glacier dynamics

The elevation change maps presented in Fig. 3 reveal a variety of decadal scale glacier change patterns. Two north-flowing clean glaciers (a and b) appear to be retreating, losing ice near their toes as most simple glacier models predict. Another large north-flowing clean glacier has experienced thinning at the transition point between a steep slope and nearly flat terrain (glacier c). The downstream "piedmont" portion of the glacier spilling onto flat terrain has not thinned as much, suggesting it is dynamically decoupled from the thinning steeper glacier portion above. The thinning pattern may also be influenced by a decrease in mass flux of the smaller confluence glacier. This would result in thinning of the ice fall at the confluence, thus strengthening the disconnect between upper and lower reaches of the glacier. The observed decoupling of the "piedmont" tongue may indicate potential for the onset of proglacial lake formation, because decreasing flow velocities and increased mass losses can induce the formation and expansion of glacial lakes under favourable topographic conditions (Thakuri et al, 2015). Modelled bed overdeepenings in this region also suggest that gently sloping thick glacier tongues of these north-flowing glaciers (including glacier c) have high potential for lake formation and enlargement (Linsbauer et al, 2016). Other glaciers terminating in nearly-flat valleys have already begun to form such lakes, which can become highly hazardous due to GLOF potential. In the Lunana region for example, the proglacial lake Lugge Tsho (located at the toe of glacier i in Figs. 2 and 3) burst on 6 October 1994 resulting in the deaths of 21 persons (Watanabe and Rothacher, 1996).

Three large south-flowing glaciers (d, e, and f) are heavily debris-covered. Modern satellite imagery viewed in Google Earth reveal melt ponds and associated ice cliffs on the surfaces of these glaciers, which can explain their irregular downwasting patterns. Recent studies have shown a disproportionately large amount of melting occurs along exposed ice cliffs compared to debris-covered regions. Supraglacial melt ponds are formed as the ice cliffs retreat, and the ponds interact with englacial conduits to enhance melting (Immerzeel et al, 2014; Reid and Brock, 2014; Sakai and Fujita, 2010). Ice cliff formation is still not well-understood, but possible mechanisms include collapse of englacial voids (initially created by drainage of melt ponds), aspect-induced differences in solar radiation, and debris slope slumping (Benn et al, 2012). A recent grid-based model of supraglacial ice cliff backwasting on debris-covered glaciers has confirmed the importance of cliffs as contributors to total mass loss of debris-covered glaciers, and shown that melt is highly variable in space, suggesting

that simple models provide inaccurate estimates of total melt volumes (Buri et al, 2015). Miles et al (2016) also showed that supraglacial ponds efficiently convey atmospheric energy to a glaciers interior, promoting the downwasting process.

Thorthormi glacier (glacier h) is a distinct example of a debris-covered calving glacier, with ice loss due to calving and thermal undercutting apparently far outweighing downwasting associated with ice cliffs and melt ponds. The largest thickness changes are occurring on the steep mid-section portion of the glacier, which may indicate a dynamic thinning response to calving as ice is lost at the glacier toe. As ice is removed from the glacier and stored in the lake, areas once covered by ice are now replaced by water, resulting in small thickness changes observed near the glacier toe. This is consistent with observations of the rapid growth of the Thorthormi lake, which is a potential GLOF hazard (Fujita et al, 2008), and suggests that ice loss is slightly underestimated by DEM differencing methods for these calving glaciers.

Glacier k has an anomalously large ice volume loss (~0.5 km$^3$), accounting for approximately 20% of the total ice volume loss of the 21 analysed glaciers. No stereo matching or georeferencing problems are apparent, and Gardelle et al (2013) show a similar large ice loss during a different timespan (1999-2011) using different elevation data (SRTM and SPOT5); thus our result is not likely due to image processing errors. It is currently unclear why this glacier has undergone such a comparatively large ice loss; however, glacier k has a large, wide accumulation area to the west (Fig. 2). One possible explanation could be that glacier thinning has caused the ice divide between glaciers k and c to shift and change position over time, thus decreasing the accumulation area and reducing the supply of ice mass for glacier k, causing a drastic reduction in volume.

### 4.3 Glacier types comparison

Profiles of ice thickness change vs. elevation show distinct thinning patterns for each glacier type (Fig 4). The clean-ice thickness change profile appears slightly positive in the accumulation zones, and thinning generally becomes greater with decreasing elevation, reaching approximately -40 m of thinning over the 32-year timespan at 5000 m elevation, then exhibiting less thinning near glacier toes. First, glacier k does not contribute to the lowest elevation bin, which results in smaller thickness change since glacier k is dominantly affecting the regional thinning profile (see Fig. S2). While some of the lower thinning rates may be due to insulating effects of more comprehensive debris-cover on glacier toes, we conclude that the primary factor is that the toes are thinner to begin with, and thus have less ice to lose. Our 1974 glacier outlines include glacier toes which were already thin at that time, and we expect thinning from 1974 onwards to be smaller near the toes. The debris-covered thickness change profile starts near zero m ice loss at 5700 m elevation, with thinning rates increasing steadily towards lower elevations, reaching around -20 m of thinning at 4200 m. The calving-glaciers thickness profile is somewhat erratic, fluctuating between -10 m and -50 m of ice loss from 6000 m down to 4400 m elevation, as several glaciers residing at different elevations have undergone significant ice loss due to calving.

Although elevation distributions of ice loss differ between glacier types, overall geodetic mass balance values for both debris-covered and clean glacier groups are similar in magnitude, with overlapping uncertainties (-0.13 ± 0.06 for clean ice and -0.19 ± 0.11 m.w.e. yr$^{-1}$ for debris-covered). This supports previous findings of similar regional averaged thinning rates

between glacier types in the Himalayas over more recent ~10-year timeframes (Kääb et al, 2012; Gardelle et al, 2013). We hypothesize that the similar magnitudes of ice loss can largely be explained by contrasting glacier hypsometries. In this region, most clean-ice glaciers have large accumulation zones, while most debris-covered glaciers have small accumulation zones. Since the debris-covered glaciers have greater proportions of ice residing at lower elevations, any given increase in temperature melts and thins a larger portion of debris-covered glacier area compared to clean-ice glacier area. As Fig. 4 illustrates, the magnitude of thinning for debris-covered glaciers is significantly less that for clean-ice glaciers, presumably due to insulating effects of the debris. However, integrating this smaller thinning across comparatively larger regions at lower elevations yields similar and even more negative mass balance values compared to the clean-ice glaciers. While these hypsometry effects are certainly not universal, further investigations are needed to determine their prevalence in other regions. Additionally, our measured geodetic thinning is influenced by both mass balance processes and ice dynamics (emergent velocities). Kääb et al (2005) showed that the large north-flowing clean-ice glaciers in Bhutan have flow velocities up to 200 m yr$^{-1}$, while south-flowing debris-covered glaciers are nearly stagnant. Thus, ice advection down-glacier is significantly greater for these clean-ice glaciers, making the apparent mass balance less negative in the ablation zones of the clean-ice glaciers as compared to the debris-covered glaciers.

Recent studies have identified relationships between glacier slope, surface velocity, and thinning rates. For example in the Langtang Himal (Nepal), zones with low surface flow velocities and low slopes tend to be associated with dynamic decay of surface features, and local accelerations in thinning for these regions correlate with development of supraglacial ice cliffs and lakes (Pellicciotti et al, 2015; Ragettli et al, 2016). We find a similar relationship in Bhutan, especially regarding glaciers d, e, and f, which have large, flat, debris-covered ablation zones, near-stagnant flow velocities (Kääb, 2005), and supraglacial ponds. As discussed in section 4.2, melt ponds, ice cliff dynamics, and englacial conduits likely play a significant role in enhancing melt for these glaciers. Additionally, longwave radiative flux change for a given temperature change is greater in regions at warmer temperatures. This may further enhance melt for lower elevation debris-covered glaciers; given that longwave atmospheric radiation is the most important heat source for melting of snow and ice (Ohmura, 2001).

Some glaciers in the region are partially debris-covered, with greater proportions of debris-covered area near glacier toes, and lower proportions of debris-covered area moving up the glacier (glaciers h and o). The mid-glacier regions with less debris-covered area exhibit greater thinning; this may be a result of enhanced ice melt due to the albedo effect of supra-glacial debris-cover that is thin enough to not provide considerable insulation effects, and the fact that bare ice melts at a faster rate than debris-covered ice at the same elevation. Modelling studies in the Khumbu region indicate that debris-covered tongues will detach from their accumulation areas in the future, leading to greater future melt rates (Shea et al, 2015; Rowan et al, 2015).

Calving glaciers in the study area have more negative mass balances compared to both types of land-terminating glaciers (both clean and debris-covered), and represent a disproportionately large amount of the total ice volume loss relative to their aerial extent. For these glaciers, large moraine-dammed lakes have formed as a result of expansion and merging of smaller

supraglacial lakes, and glacial meltwater is effectively stored adjacent to glacier termini (Basnett et al, 2013). As changing climate increases glacier melt, the resulting lakes interact with remaining ice to further enhance melt through thermal undercutting processes independent of climate (Sakai et al, 2009; Thompson et al, 2012). This positive feedback mechanism has important implications for future hazard and water resource issues, especially for glaciers terminating in flat valleys with potential lake-forming topographies. Gardelle (2011) estimated that in the eastern HKH (India, Nepal, and Bhutan) glacial lakes have grown continuously between 1990 and 2009 by 20% to 65%. Thus, these glacier-lake systems not only represent GLOF hazards, but will likely play a key role in the Himalayan ice mass budget during the coming decades.

## 5 Conclusions

We applied a new DEM extraction pipeline toward Hexagon spy satellite imagery and ASTER data to compute glacier thickness changes over a multi-decadal timescale across a large glacierized area (~365 km$^2$) in the eastern Himalayas. The consistency of the DEM extraction method provided high geolocational accuracy and minimized elevation biases when differencing the DEMs. In addition, the long timespan (1974-2006) allowed for effective separation of glacier change from noise inherent in the remote sensing methods. Results provide insight into the complex dynamics of glaciers in the monsoonal Himalayas, and highlight similarities and differences in the decadal responses of clean, debris-covered, and calving glaciers. Though regional thinning and ice loss is apparent, individual glacier dynamics vary widely depending on elevation, hypsometry, extent and thickness of debris, and potential for calving in proglacial lakes. Both clean and debris-covered glaciers show similar negative geodetic mass balances, while lake-terminating glaciers have geodetic mass balances more negative than land-terminating glaciers. The more negative mass balances of lake-terminating glaciers suggest that calving and thermal undercutting are important mechanisms contributing to ice loss in the region, and highlights the rapidly growing hazard potential of GLOFs in the monsoonal Himalayas. Overall, these results enhance understanding regarding potential glacier contribution to sea-level rise, impact on hydrological resources, and hazard potential for high mountain regions and downstream populations in Asia.

*Acknowledgements.* This work was funded by NSF Grants 1256551 and 1304397 to SR, and a Rocky Mountain NASA Space Grant Consortium (RMNSGC) fellowship to JM. JMS acknowledges support by the NSF Grant EAR-1304351 and by the Lamont Climate Center. We thank the USGS, NASA, and Japan Space Systems for providing access to Hexagon and ASTER data, and gratefully acknowledge Barry Bickmore and Jani Radebaugh for constructive comments on the manuscript.

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

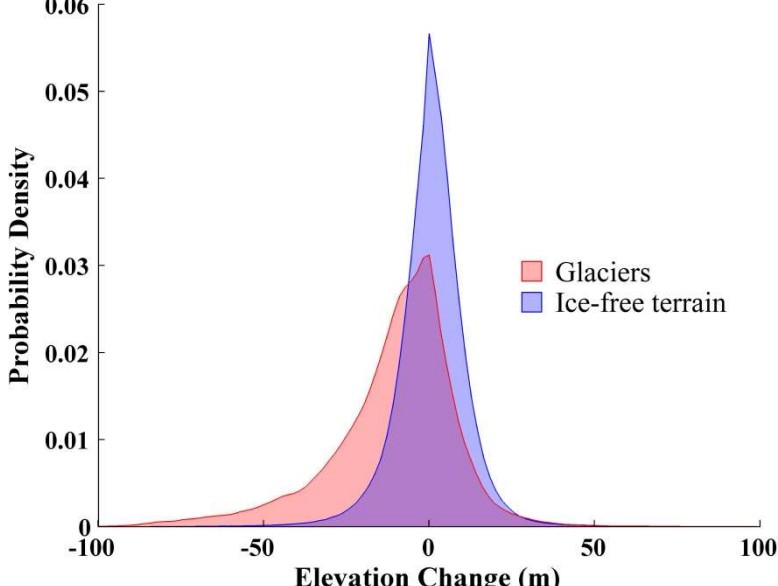

**Figure 1. Probability density distributions for all pixels in the 2006 minus 1974 elevation change maps obtained via DEM differencing, separated into glacial ice terrain and surrounding ice-free terrain groups. The glacial terrain distribution has mean = -10.9 m, median = -7.3 m, and σ = 19.7 m. By comparison, the ice-free terrain distribution has mean = 0.7 m, median = 0.9 m,**
5  **and σ = 10.9 m. Non-zero elevation changes in the ice-free terrain distribution are likely caused by a combination of actual changes such as landslides, along with intrinsic elevation error associated with stereo photogrammetric methods used to create the DEMs.**

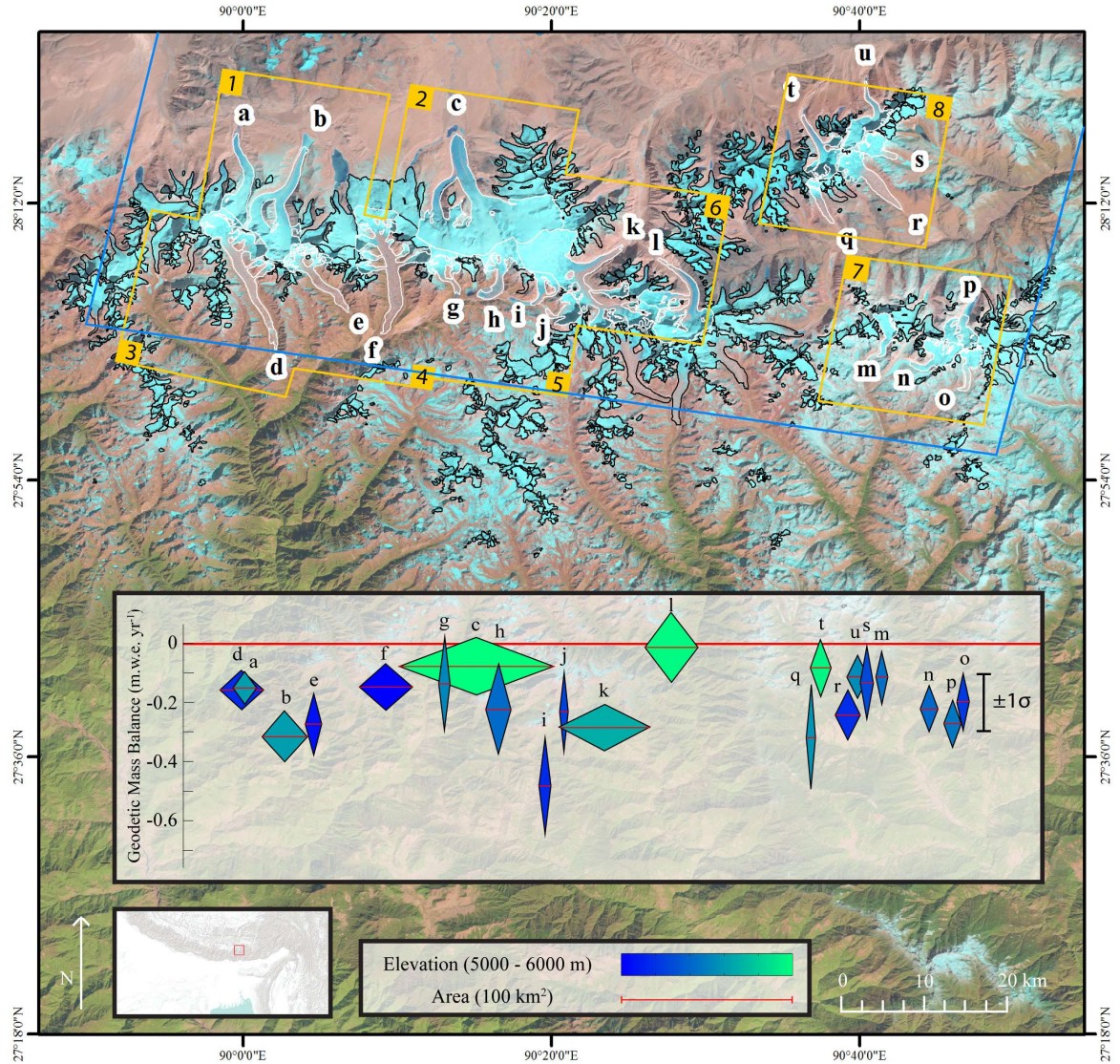

**Figure 2. Landsat 8 image showing study region located in the eastern Himalayas and Tibetan Plateau. Black outlines identify all glaciers in the region, while white outlines denote glaciers used in this study, identified by letters a-u. Glacier outlines were downloaded from the ICIMOD mountain geoportal. Orange outlines indicated extent of extracted 1974 Hexagon DEMs; blue line indicates extent of the 2006 ASTER DEM. Inset: Annual geodetic mass balances for selected glaciers during the 1974 to 2006 period (2006 ASTER DEM minus 1974 Hexagon DEMs), where each diamond represents a glacier. Central red lines are geodetic mass balances for each glacier in m.w.e. $yr^{-1}$ (meters water equivalent per year). Diamond widths are proportional to total glacier area, heights indicate ±1 standard error (SEM) uncertainty, and colors specify mean glacier elevations. Thick red line indicates zero change.**

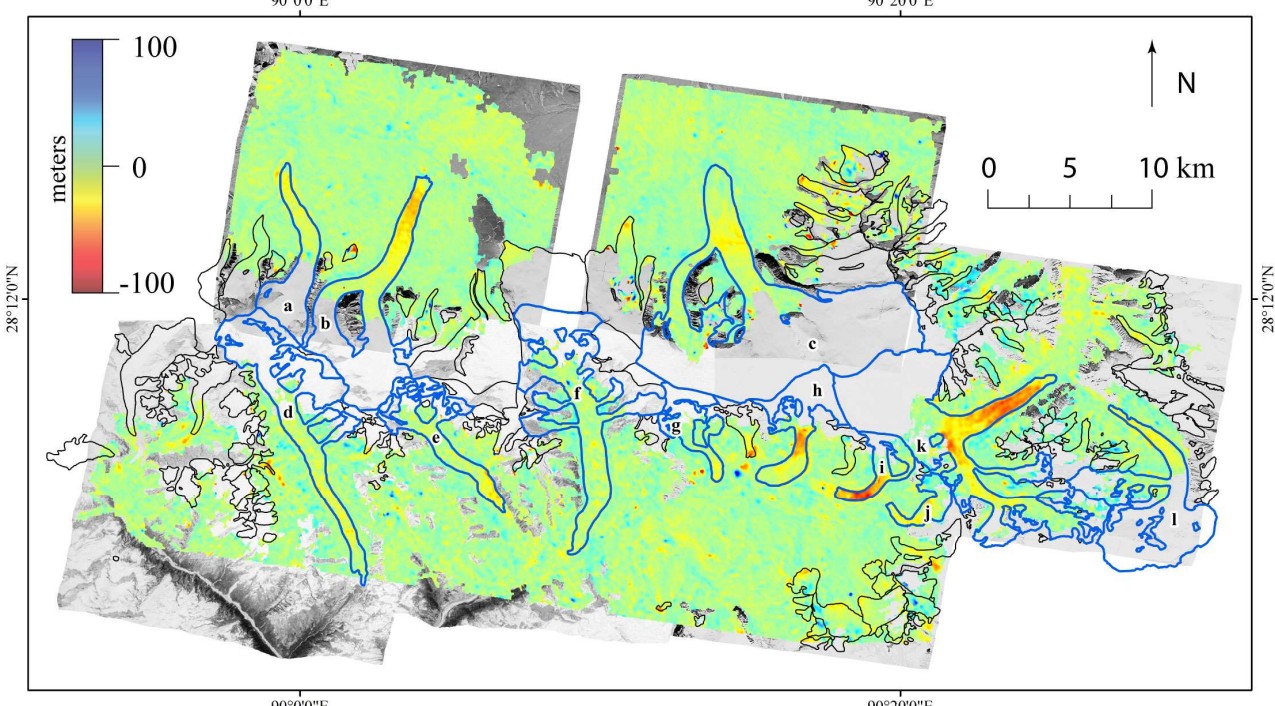

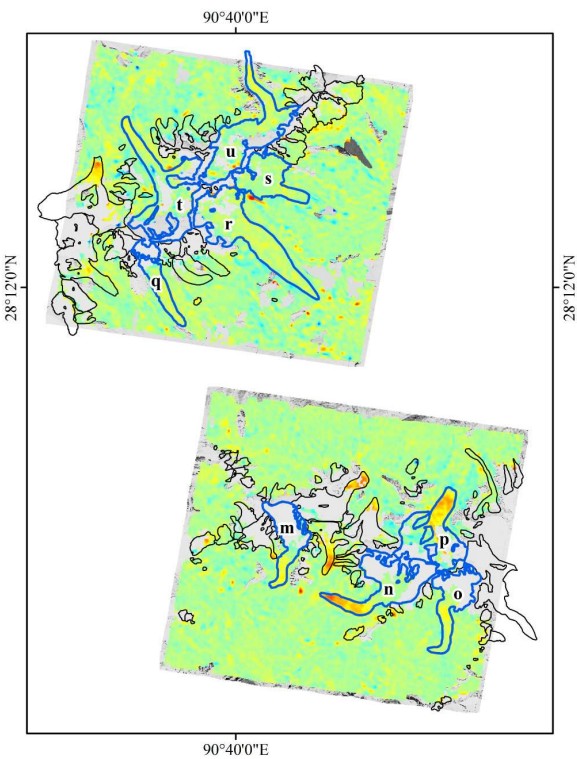

**Figure 3. Elevation change maps for 2006 minus 1974. Blue outlines denote glaciers used in this study, identified by letters a-u. Note regions of missing data in glacier accumulation zones, where the stereo matching algorithm failed due to poor radiometric contrast and oversaturation caused by snow cover. Glaciers a and b exhibit thinning near their toes, while glacier c is thinning at the transition point between a steep slope and nearly flat terrain. Three large debris-covered glaciers (d-f) show somewhat irregular patterns of thinning due to downwasting.  Glaciers g-j (located in the Lunana region of Bhutan where a 1994 fatal GLOF event occurred) show significant thinning and retreating of glacier toes, which have contributed to the growth of unstable moraine-dammed proglacial lakes (glaciers g, h, and i are classified as calving glaciers in this study).  Glacier k shows the greatest ice volume loss in the study region. Glaciers m-p are located in eastern Bhutan, and also show significant downwasting and retreat. Glaciers q-u are the most northeastern, mostly debris-covered, and show a moderate rate of thinning.**

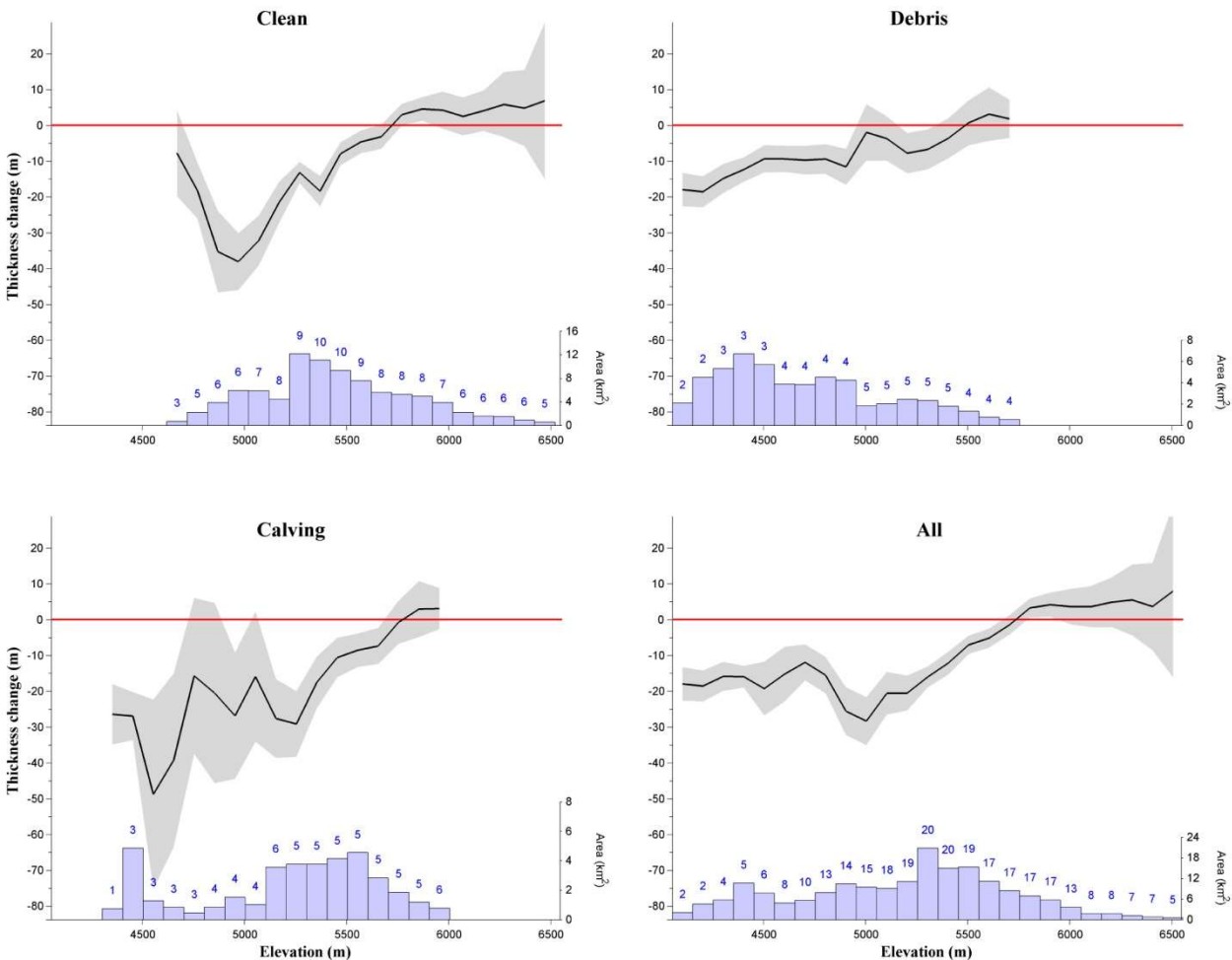

**Figure 4. Ice thickness change profiles for clean ice, debris covered, calving, and all glaciers (rates of change can be obtained by dividing values on the vertical axis by 32 years). Thickness change pixels are separated into 100 m bands; black lines are the mean, and the grey shaded regions represent the standard error of the mean estimated as $\sigma_z/\sqrt{n_z}$ , where $\sigma_z$ is the standard deviation of elevation change for each band and $n_z$ is calculated using Eq. 5. The glacier area (km$^2$) contained in each band is indicated by the blue histogram bars, and the number of glaciers contributing to each elevation band is shown by the blue number above each bin. Note that the histogram bars do not include extrapolated data.**