# Peer review of "Quantifying ice loss in the eastern Himalayas since 1974 using declassified spy satellite imagery"

_The Cryosphere, 2016_

## Referee Comment (RC1) · T. Nuimura (Referee) · 9 May 2016

General comments

This manuscript addresses glacier variation over long period (1974–2006). The consistent procedure for generating DEMs use with HEXIMAP enables DEM differentiation with high accuracy, and overcomes procedure-dependent error. The estimated regional mass budget shows little discrepancy with previous studies. However, authors explanation about the discrepancy come from difference of analysed time span (previous studies only cover last decade) is reasonable.

Specific comments
P5 L9-10: Could you show the rate of estimated thickness change in each elevation band in Figure? For example, stack histogram in Fig. 4 might be better (extrapolated data with another color on blue histogram bar).

P6 L28: Does "standard error (SEM)" mean "standard error of the mean (SEM)" or "standard error (SE)"?

P8 L12-13: The geodetic mass balance by Kääb et al. (2012) also includes East Nepal. Strictly speaking, elevation difference around Bhutan is more negative. In Gardelle et al. (2013), they re-calculated it as -0.52±0.16 in Table 5. The value should be used here.

P10 L13: Fig.4 is appropreate figure for checking summary of glacier variation tendency for each glacier type. However, such aggregation of each data make unclear each glacier characteristics. Is it possible to add ice thickness change profile of each glacier as colored lines into Fig.4? If it makes Fig. 4 ambiguous, please add it as another new figure.

P10 L13–20: You have mentioned the three possibilities of the small lowering elevation bands (about 4600-4800 m) here. I recommend to investigate the reason by checking individual glaciers. Number of glaciers in these elevation bands (about 4600–4800 m) are 2 and 6 (from Fig. 4), so it is not laborious work. I guess your second possibility is correct.

P11 L1–20: As I commented before, figure about ice thickness change profile for each glacier should be add. It make easy to understand discussion here.

Figure 3: Description about elevation change for each glaciers (Glaciers a and b ...) should be moved to main text.

typing errors

P9 L17, P15 L29: Correctly, his family name is not Watanbe but Watanabe. It is erratum by the journal 'Mountain Research and Development'.

---

## Referee Comment (RC2) · T. Pieczonka (Referee) · 10 May 2016

**Review TC-2016-48**

**General Comments**

The potential of declassified optical stereo satellite imagery for glacier change detection is undisputed. Several studies quantified glacier volume changes using declassified Corona and Hexagon data, mainly focusing on small basins or individual glaciers only. For the processing nearly all of the studies used commercial software packages. From this point of view the study of Maurer is highly welcomed as it proposed an independent workflow for DEM generation. Moreover, the study of Maurer closes a gap of knowledge in terms of region-wide glacier mass balance investigations in Bhutan using declassified imagery.

The workflow of DEM generation has already been published in the ISPRS Journal of Photogrammetry and Remote Sensing (Maurer and Rupper, 2015). This paper builds upon this previous paper and presents an application in the field of glacier volume change assessment. All in all it is a nice paper and worth of prompt publishing.

**Specific Comments**

P 3, L 29. How were the blocks selected? Were they defined with respect to the glacier extent?

P 4, L2. "Points located on unstable terrain were excluded" – Based on ICIMOD glacier outlines?

P 4, L 21. Hexigon -> Hexagon

P 4, L 29. I am not familiar with glacier elevation changes in Bhutan. Comparing with the Everest Region 100 m seems to be suitable. However, the strong elevation change for glacier "k" and "i" might also justify a threshold of 150 or 200 m.

P 5, L 2. "Greatest accuracy" – What does it mean? Did you use a slope threshold to correct the outlines with regard to steep parts in the accumulation regions which have erroneously been delineated as part of the glacier? The delineation of debris-covered glacier tongues is often ambiguous. How did you cope with uncertainties in these regions (e.g. glacier "f" in Figure 3 shows a strong glacier thinning at the end of the tongue outside the glacier outline)?

P 5, L9-11. Data gaps in the accumulation regions are characteristic for DEMs generated using optical satellite imagery. Closing these gaps is indispensable for glacier mass balance calculations. Several approaches can be found in the literature, e.g. based on TINs (Surazakov and Aizen, 2006) or using the regional mean/median for individual elevation bands. Other studies assumed no change in the accumulation regions replacing missing data values by zero (Pieczonka and Bolch, 2013). Using the regional mean the authors assume a similar behavior for all glaciers of the same type. However, this must not necessarily be true. Taking this into account the proposed method allows the calculation of regional glacier volume changes but might not be suitable to calculate glacier volume changes for individual glaciers (e.g. P 10, L 4). Can you provide a difference image after extrapolation to judge the meaningfulness of the approach? You could also add a figure showing all three stages for an example glacier (difference image before hole interpolation, after hole interpolation, after extrapolation).

P 8, L12. The results from this study are not comparable to the results of Kääb et al. (2012) and Gardelle et al. (2013) due to different time periods. The authors should add other references when

writing "is comparable to other estimates derived from remote sensing […]". Compared to the Everest Region (Bolch et al., 2011) the regional mass budget is less negative. Zemp at al. (2009) give annual mass balances based on in-situ observations on a broader scale. For Southeast Asia they found a moderate mass loss until 1995 with a subsequent acceleration since 1996, reflecting the higher mass loss found by Kääb et al. (2009) and Gardelle et al. (2012) for the last decade.

P 9, L 20. What is the source for the information regarding melt ponds? Did the authors use Google Earth for a visual inspection of the glaciers?

P 10, L 15 and L 20. In my opinion it is a decrease from -35 to -10 m in line 15 and an increase from 0 to -10 m in line 20 as the values are related to glacier thinning.

P 11, L 7. "Low slope and low surface velocity" – This is imprecise and some further information is needed.

P 11, L 9. "Near stagnant flow velocities" – This statement is not supported by any figure or by a reference.

P 11, L 14. "heavy debris" and "lighter debris" – What does it mean in terms of absolute debris thickness?

Figure 2. 8 different Hexagon DEMs are mentioned. Figure 2 shows only 3 polygons. This needs to be changed to 8 polygons to get the link between Table S1 and Figure 2.

Figure 3. The chosen color for the investigated glaciers is unfavorable in comparison to the bright background color. A different color for the outlines would improve the readability of the figure significantly (in particular for the glaciers "q"-"u").

Figure 3. The current visualization using a continuous color coding shows an almost perfect fit between both models but conceals uncertainties in the data. The authors should use classes instead of a continuous color coding in order to allow a better distinction between areas of higher and lower deviations from zero, in particular for the stable terrain.

Figure 3. Glacier "a" shows a strong surface lowering in the middle part of the tongue followed by sudden elevation uplift at the end of the tongue. At the same time there is a strong thinning right next to the outline not mapped as a glacier.

Figure 3. Considering glacier "k" and "i" I would expect a surface lowering of more than 100 m. The threshold of 100 m used to exclude outliers might be too low. A value of 150 or 200 m might be more suitable, in particular for the ablation regions. Can the author provide a difference image before gap interpolation in the supplements?

Table S2. Area uncertainties for 1974 are missing.

---

## Referee Comment (RC3) · J.M. Shea (Referee) · 23 May 2016

Review - Maurer and Rupper, TCD

Using declassified spy satellite imagery (Hexagon and Corona) and recent ASTER imagery, Maurer et al calculate glacier mass changes for 21 glaciers in eastern Bhutan, from the 1974 to 2006. The methods and assumptions appear to be sufficient, errors in the analysis are well-documented, and the results are interesting and highly relevant. The paper is also well-referenced, very well-written, and essentially free from grammatical/structural/organizational errors.

Aside from the comments by the other reviewers (Pieczonka and Nuimura), I can only

add a few general points that the authors might wish to address:

1) A very brief outline of the Hexagon/Corona pipeline (Maurer and Rupper, 2015) would be helpful

2) Why are only 21 glaciers studied? And what are the impacts of the 30% coverage threshold? Previous geodetic studies (e.g. Gardelle et al., 2013, Bolch et al., 2011) consider the entire glacierized area that is covered within a region. Using only 21 glaciers (and only those larger than 3 km2) and replacing potentially large missing areas with the regional mean for a specific glacier type and elevation band could result in biased regional estimates of glacier mass change that are not comparable to previous studies.

3) How is the ELA defined in this study (it first appears on P10L29)? Strictly speaking, this is typically taken from surface mass balance measurements. While the elevation that divides geodetic mass gain and loss would be related to the ELA, I am not sure that it can be used as an ELA substitute (though I would be interested to hear otherwise).

Specific comments: P2L20: are these annual or seasonal streamflow contributions?

P4L20: Define DN.

P10L25: What data support the conclusion that debris-covered glaciers melt at the same rate as clean-ice glaciers? If this is overall mass balance rates than it should be specified. Figure 4 clearly shows that melt rates at debris-covered glaciers are lower than those observed on clean ice glaciers for the same elevation band, and this is later referenced by the authors on P10L28.

P11L15: Debris cover will almost always get thinner moving up-glacier. The greater thinning rates observed at the transition between debris-covered and debris-free zones is due in part to enhanced melt rates under thin debris cover but also due to the simple fact that bare ice will melt at a faster rate than debris-covered ice at the same elevation. Modelling studies in the Khumbu region (Shea et al., 2015; Rowan et al., 2015) both

indicate that debris-covered tongues will detach from their accumulation areas in the future, leading to greater future melt rates.
* * *

---

## Author Comment (AC3) · 1 Jul 2016

Using declassified spy satellite imagery (Hexagon and Corona) and recent ASTER imagery, Maurer et al calculate glacier mass changes for 21 glaciers in eastern Bhutan, from the 1974 to 2006. The methods and assumptions appear to be sufficient, errors in the analysis are well-documented, and the results are interesting and highly relevant. The paper is also well-referenced, very well-written, and essentially free from grammatical/structural/organizational errors.

Thank you for your helpful comments on the manuscript. We have addressed all of your concerns below, and feel they have improved the paper considerably.

Aside from the comments by the other reviewers (Pieczonka and Nuimura), I can only add a few general points that the authors might wish to address:

1) A very brief outline of the Hexagon/Corona pipeline (Maurer and Rupper, 2015) would be helpful.

A brief summary of the process is now included in section 2.1 on P4.

2) Why are only 21 glaciers studied? And what are the impacts of the 30% coverage threshold? Previous geodetic studies (e.g. Gardelle et al., 2013, Bolch et al., 2011) consider the entire glacierized area that is covered within a region. Using only 21 glaciers (and only those larger than 3 km2) and replacing potentially large missing areas with the regional mean for a specific glacier type and elevation band could result in biased regional estimates of glacier mass change that are not comparable to previous studies.

Unfortunately, low radiometric contrast, cloud cover, and spatially correlated noise/error in the DEMs prevent accurate calculation of changes for all glaciers in the region. While this does limit direct comparison to other previous studies (this among other things, such as different timespans covered), we feel these 21 large glaciers give a good regional picture of thickness changes over the 3 decade timespan. We have updated the discussion to more accurately reflect these facts in the paragraph starting on P6 L14.

We agree that replacing potentially large missing areas with regional means may result in biased regional estimates, and now include results using both the regional extrapolation method vs. assuming zero change for missing data (Table S3). Addition discussion of the observed impacts of extrapolation vs. assuming zero change is also included on P6 L3 and P9 L6.

3) How is the ELA defined in this study (it first appears on P10L29)? Strictly speaking, this is typically taken from surface mass balance measurements. While the elevation that divides geodetic mass gain and loss would be related to the ELA, I am not sure that it can be used as an ELA substitute (though I would be interested to hear otherwise).

The term "ELA" was being used too loosely here, and instead we substitute the term "glacier hypsometry." Updated on P11 L20.

Specific comments:

P2L20: are these annual or seasonal streamflow contributions?

These are seasonal, their samples (from which their streamflow contributions were derived) were collected during September (post-monsoon). We have updated the text to make this important distinction clear.

The text has been updated on P2 L19.

P4L20: Define DN.

DN = Digital Numbers, these are simply the pixel values in Landsat and ASTER images before being converted to reflectance or radiance. Updated on P4 L28.

P10L25: What data support the conclusion that debris-covered glaciers melt at the same rate as clean-ice glaciers? If this is overall mass balance rates than it should be specified. Figure 4 clearly shows that melt rates at debris-covered glaciers are lower than those observed on clean ice glaciers for the same elevation band, and this is later referenced by the authors on P10L28.

We now clarify at the beginning of the section, that although elevation distributions of ice loss differ between clean-ice and debris-covered glacier groups, overall geodetic mass balance values are similar in magnitude. Updated on P11 L16.

P11L15: Debris cover will almost always get thinner moving up-glacier. The greater thinning rates observed at the transition between debris-covered and debris-free zones is due in part to enhanced melt rates under thin debris cover but also due to the simple fact that bare ice will melt at a faster rate than debris-covered ice at the same elevation. Modelling studies in the Khumbu region (Shea et al., 2015; Rowan et al., 2015) both indicate that debris-covered tongues will detach from their accumulation areas in the future, leading to greater future melt rates.

We now include this information and accompanying references in the text on P12 L12.

---

## Author Response (AR1)

General comments

This manuscript addresses glacier variation over long period (1974–2006). The consistent procedure for generating DEMs use with HEXIMAP enables DEM differentiation with high accuracy, and overcomes procedure-dependent error. The estimated regional mass budget shows little discrepancy with previous studies. However, authors explanation about the discrepancy come from difference of analysed time span (previous studies only cover last decade) is reasonable.

Thank you for your thoughtful consideration of our manuscript. We have carefully considered your comments, and address all of them directly below.

Note: all page and line numbers refer to the final manuscript version (not the marked-up manuscript version with track changes).

Specific comments

P5 L9-10: Could you show the rate of estimated thickness change in each elevation band in Figure? For example, stack histogram in Fig. 4 might be better (extrapolated data with another color on blue histogram bar).

The rate of change is a simple scalar of the thickness change (i.e. thickness change divided by 32 years). We would like to keep this figure as simple as possible, and thus avoid adding extra

information in the elevation bands. Accordingly, we have left the figure as is, and instead notify the reader that in order to get the rate of change, values on the vertical axis can be divided by the timespan of 32 years.

To provide additional visualization of individual glacier changes and processing steps, we now include additional supplemental figures – 1) Fig S2 showing individual glacier profiles, and 2) Fig S3-S5 which show the processing steps for each glacier, including areas with interpolated data.

P6 L28: Does "standard error (SEM)" mean "standard error of the mean (SEM)" or "standard error (SE)"?

This was a typographical error, we have corrected it, and now state "standard error of the mean" on P7 L30.

P8 L12-13: The geodetic mass balance by Kääb et al. (2012) also includes East Nepal. Strictly speaking, elevation difference around Bhutan is more negative. In Gardelle et al. (2013), they re-calculated it as -0.52_0.16 in Table 5. The value should be used here.

We fixed this to show updated values from Table 5 in Gardell et al (2013) as suggested on P8 L30

P10 L13: Fig.4 is appropreate figure for checking summary of glacier variation tendency for each glacier type. However, such aggregation of each data make unclear each glacier characteristics. Is it possible to add ice thickness change profile of each glacier as colored lines into Fig.4? If it makes Fig. 4 ambiguous, please add it as another new figure.

A new figure (Figure S2) is now included in the supplement showing thickness changes for individual glaciers.

P10 L13–20: You have mentioned the three possibilities of the small lowering elevation bands (about 4600-4800 m) here. I recommend to investigate the reason by checking individual glaciers. Number of glaciers in these elevation bands (about 4600–4800 m) are 2 and 6 (from Fig. 4), so it is not laborious work. I guess your second possibility is correct.

As recommended, we carefully inspected the individual glaciers contributing to the lower elevation bins, using both Figure S2, the Hexagon and ASTER satellite imagery, as well as high resolution Google Earth imagery.

For clean-ice glaciers:

As noted in the text (and which can be seen in figure S2), glacier k is the dominant factor affecting the clean-ice profile in Figure 4.  Since glacier k does not contribute to the lowest elevation bin, the bin exhibits an apparent smaller thickness change.

On closer inspection of glacier toes, we observe that several glacier toes appear darker toward their termini, which we interpret as increased amount of debris cover (glacier s and glacier o for example). The insulating effects of debris-cover likely contribute somewhat to the observed pattern, but based on our analysis, we conclude that the primary factor explaining this phenomenon is that glacier toes are thin to begin with, and thus have less ice to lose. Our 1974 glacier outlines include glacier toes which were already thin at that time, and we expect thinning from that point in time onwards to be smaller near the termini.

For debris-covered glaciers

Polygon glacier outlines have accuracy problems near debris-covered glacier toes. This is a well-known problem, as heavy debris-cover is indistinguishable from surrounding terrain. Unfortunately, without field measures of debris-thickness we find it impossible to back out the relative contributions of insulation effects vs. inaccurate glacier polygons at debris-covered glacier toes. However, we suggest that future work using SAR velocities may be able to address this problem.

For calving glaciers:

As mentioned in the text, meltwater is effectively stored adjacent to glacier termini in proglacial lakes, making the thickness change appear smaller due to the filling effect of the lake.

We have updated the text to better reflect these findings in section 4.3 on P11.

P11 L1–20: As I commented before, figure about ice thickness change profile for each glacier should be add. It make easy to understand discussion here.

We agree, and now include such a figure (Figure S2).

Figure 3: Description about elevation change for each glaciers (Glaciers a and b ...) should be moved to main text.

The discussion has been moved to main text as suggested.

typing errors

P9 L17, P15 L29: Correctly, his family name is notWatanbe butWatanabe. It is erratum by the journal 'Mountain Research and Development'.

Thank you, we have updated his family name.

**Review TC-2016-48**

**General Comments**

The potential of declassified optical stereo satellite imagery for glacier change detection is undisputed. Several studies quantified glacier volume changes using declassified Corona and Hexagon data, mainly focusing on small basins or individual glaciers only. For the processing nearly all of the studies used commercial software packages. From this point of view the study of Maurer is highly welcomed as it proposed an independent workflow for DEM generation. Moreover, the study of Maurer closes a gap of knowledge in terms of region-wide glacier mass balance investigations in Bhutan using declassified imagery.

The workflow of DEM generation has already been published in the ISPRS Journal of Photogrammetry and Remote Sensing (Maurer and Rupper, 2015). This paper builds upon this previous paper and presents an application in the field of glacier volume change assessment. All in all it is a nice paper and worth of prompt publishing.

Thank you for your positive comments on the approach and results. We also appreciate your careful consideration of the manuscript and your detailed comments/criticisms.  We have carefully considered your comments, and address all of them directly below.

Note: all page and line numbers refer to the final manuscript version (not the marked-up manuscript version with track changes).

**Specific Comments**

P 3, L 29. How were the blocks selected? Were they defined with respect to the glacier extent?

Blocks were selected to maximize coverage of large glaciers across the region, and avoid regions with cloud cover. We have clarified this point in the text on P3, L30.

P 4, L2. "Points located on unstable terrain were excluded" – Based on ICIMOD glacier outlines?

The ICIMOD glacier outlines were used to exclude glacier terrain during optimization.  The glacier outlines were first converted to a raster binary mask at the spatial resolution of the reference DEM used during optimization (SRTM in this case).  Next, dilation (a morphological operation which adds pixels to edge boundaries) was used to slightly enlarge the glacier boundaries in the raster mask, which helped to eliminate any unstable glacier pixels not quite contained by the ICIMOD outlines, as well as possibly unstable moraines.

Additionally, in the optimization routine, any elevation change pixels outside of 3 standard deviations are excluded at each iteration, which effectively eliminates other unknown sources of large error during optimization.

We have updated the text in section 2.1 on P3-4 to more precisely explain our methods.

P 4, L 21. Hexigon -> Hexagon

Thank you for correcting this typographical error.

P 4, L 29. I am not familiar with glacier elevation changes in Bhutan. Comparing with the Everest Region 100 m seems to be suitable. However, the strong elevation change for glacier "k" and "i" might also justify a threshold of 150 or 200 m.

After checking thickness changes for the two glaciers with strongest changes (glaciers i and k), we find that for glacier i, the largest magnitude thickness change is -104 m (negative meaning the glacier has thinned), and only 24 pixels out of 6939 are more negative than -100 m (0.4%). For glacier k, the largest magnitude thickness change is -93 m, thus 0 pixels out of 57182 are more negative than -100. Given the uncertainties in the elevation change measurements, we find that the few excluded pixels in glacier i exhibiting thickness changes slightly more negative than -100 meters do not significantly affect results.

P 5, L 2. "Greatest accuracy" – What does it mean? Did you use a slope threshold to correct the outlines with regard to steep parts in the accumulation regions which have erroneously been delineated as part of the glacier? The delineation of debris-covered glacier tongues is often ambiguous. How did you cope with uncertainties in these regions (e.g. glacier "f" in Figure 3 shows a strong glacier thinning at the end of the tongue outside the glacier outline)?

At that time we inspected glacier polygon outlines from both the Randolph Glacier Inventory and ICIMOD. We found that polygons from the Randolph Glacier inventory had significant georeferencing errors in the Bhutan region, while those from ICIMOD appeared more detailed and accurate, based on visual inspection after overlying the outlines on high resolution Google Earth imagery.  Recently these exact same ICIMOD outlines have been incorporated into the GLIMS database, but at that time they were not. We have updated the text on P5 L10.

As suggested, we have updated our workflow to include a slope threshold, to correct glacier areas by removing erroneously delineated parts (i.e nunataks, rock cliffs, etc.).  We found that a threshold of 45 degrees was optimal, and have updated our results with this correction included. The text has also been updated on P5 L 13.

As noted in our uncertainties section, we expect relative vertical errors to occur between the Hexagon and ASTER DEMs, i.e. apparent elevation changes where no actual changes have occurred. This is due to different sensor characteristics, viewing geometries, sun position, cloud cover and atmospheric conditions, etc. between Hexagon and ASTER imagery. It is entirely possible these errors occur near a glacier toe, making it appear as though the change is part of the glacier, when in fact it is not. Further complicating this are actual non-ice elevation changes such a landslides and glacial-lake outburst floods, which can be triggered as glaciers recede and alter stress regimes along valley walls and moraine ridges, exposing unstable slopes, and reorganizing large volumes of unconsolidated sediment (e.g. Richardson, Shaun D., and John M. Reynolds. "An overview of glacial hazards in the Himalayas." Quaternary International 65 (2000): 31-47). Therefore, we base our glacier outlines primarily on the satellite imagery, with the thickness change maps as a secondary source.  We also include a new supplementary figure, S6, which illustrates some unstable moraines which have since collapsed in the region, leading to non-ice elevation changes near glaciers. The associated text has been updated in the paragraph starting on P5 L16.

We agree regarding the common problem of ambiguous debris-covered glacier tongues. We have manually adjusted the ICIMOD glacier outlines to capture glacier extents to the best of our ability, however it is entirely possible (indeed likely) that some debris-covered toes are slightly inaccurate, as is the case for every previous study using remote sensing methods to study debris-covered glaciers. Unfortunately there is no foreseeable way of eliminating ambiguity regarding the separation of inaccurate glacier outlines, DEM errors, or reorganization of unconsolidated sediment, especially regarding glacier extents during the 1970's. We now discuss this problem in the text, and suggest a possible route for future studies to better delineate debris-covered glaciers (P5 L16).

P 5, L9-11. Data gaps in the accumulation regions are characteristic for DEMs generated using optical satellite imagery. Closing these gaps is indispensable for glacier mass balance calculations. Several approaches can be found in the literature, e.g. based on TINs (Surazakov and Aizen, 2006) or using the regional mean/median for individual elevation bands. Other studies assumed no change in the accumulation regions replacing missing data values by zero (Pieczonka and Bolch, 2013). Using the regional mean the authors assume a similar behavior for all glaciers of the same type. However, this must not necessarily be true. Taking this into account the proposed method allows the calculation of regional glacier volume changes but might not be suitable to calculate glacier volume changes for individual glaciers (e.g. P 10, L 4). Can you provide a difference image after extrapolation to judge the meaningfulness of the approach? You could also add a figure showing all three stages for an example glacier (difference image before hole interpolation, after hole interpolation, after extrapolation).

We now include results using both gap-filling methods (regional extrapolation for individual elevation bands and assuming zero change) in order to facilitate comparison in Table S3. In addition, 3 new supplementary figures are included which show each glacier at each processing stage (Fig S3-S5).

The text has been updated in the paragraph starting on P6 L3.

P 8, L12. The results from this study are not comparable to the results of Kääb et al. (2012) and Gardelle et al. (2013) due to different time periods. The authors should add other references when writing "is comparable to other estimates derived from remote sensing […]". Compared to the Everest Region (Bolch et al., 2011) the regional mass budget is less negative. Zemp at al. (2009) give annual mass balances based on in-situ observations on a broader scale. For Southeast Asia they found a moderate mass loss until 1995 with a subsequent acceleration since 1996, reflecting the higher mass loss found by Kääb et al. (2009) and Gardelle et al. (2012) for the last decade.

We agree that the wording of this statement is inaccurate, as different time periods prevent direct comparison. The text has been updated accordingly in section 4.1 on P8-9, and results from the other studies mentioned have been added.

P 9, L 20. What is the source for the information regarding melt ponds? Did the authors use Google Earth for a visual inspection of the glaciers?

The source of the information was imagery viewed in Google Earth (updated on P10 L10).

P 10, L 15 and L 20. In my opinion it is a decrease from -35 to -10 m in line 15 and an increase from 0 to -10 m in line 20 as the values are related to glacier thinning.

We agree the wording was confusing, and have updated to more clearly describe the profiles in section 4.3 on P11.

P 11, L 7. "Low slope and low surface velocity" – This is imprecise and some further information is needed.

The statement has been updated to provide more information on P11 L33.

P 11, L 9. "Near stagnant flow velocities" – This statement is not supported by any figure or by a reference.

The reference to (Kaab, 2005) has been added on P12 L3.

P 11, L 14. "heavy debris" and "lighter debris" – What does it mean in terms of absolute debris thickness?

We mean that certain regions appear to have more or less area covered by debris, based on visual inspection of the satellite imagery. Darker = relatively more area covered by debris, lighter = relatively more area covered by ice.

The text has been updated to more precisely define our meaning on P12 L9.

Figure 2. 8 different Hexagon DEMs are mentioned. Figure 2 shows only 3 polygons. This needs to be changed to 8 polygons to get the link between Table S1 and Figure 2.

Labels for the 8 different Hexagon DEM regions have been added to Figure 2.

Figure 3. The chosen color for the investigated glaciers is unfavorable in comparison to the bright background color. A different color for the outlines would improve the readability of the figure significantly (in particular for the glaciers "q"-"u").

We agree, and have changed the color of investigated glacier outlines from white to blue.

Figure 3. The current visualization using a continuous color coding shows an almost perfect fit between both models but conceals uncertainties in the data. The authors should use classes instead of a continuous color coding in order to allow a better distinction between areas of higher and lower deviations from zero, in particular for the stable terrain.

We now include a version of Figure 3 with classes in the supplement (Fig S7). However, we feel that a version with continuous color coding is essential for clearly illustrating the various spatial patterns of thinning with the most detail. Thus, we have retained the continuous version, but have improved it by enhancing the contrast to better highlight the elevation differences.

Figure 3. Glacier "a" shows a strong surface lowering in the middle part of the tongue followed by sudden elevation uplift at the end of the tongue. At the same time there is a strong thinning right next to the outline not mapped as a glacier.

On close inspection, we find a lateral moraine ridge has collapsed along the west margin of the glacier tongue, which likely caused the apparent elevation change there (Fig S6).

As Fig. S6 shows, the polygon does contain ice at the end of the tongue.  We hypothesize that the apparent elevation uplift may be due to emergent velocities associated with glacier dynamics.

Figure 3. Considering glacier "k" and "i" I would expect a surface lowering of more than 100 m. The threshold of 100 m used to exclude outliers might be too low. A value of 150 or 200 m might be more suitable, in particular for the ablation regions. Can the author provide a difference image before gap interpolation in the supplements?

See previous comment regarding P4, L 29.  We also now include Figs S3-S5 to show difference images at various stages, including before and after gap interpolation.

Table S2. Area uncertainties for 1974 are missing.

The area uncertainties have been added.

Using declassified spy satellite imagery (Hexagon and Corona) and recent ASTER imagery,
Maurer et al calculate glacier mass changes for 21 glaciers in eastern Bhutan,
from the 1974 to 2006. The methods and assumptions appear to be sufficient, errors
in the analysis are well-documented, and the results are interesting and highly relevant.
The paper is also well-referenced, very well-written, and essentially free from
grammatical/structural/organizational errors.

Thank you for your helpful comments on the manuscript. We have addressed all of your concerns
below, and feel they have improved the paper considerably.

Note: all page and line numbers refer to the final manuscript version (not the marked-up
manuscript version with track changes).

Aside from the comments by the other reviewers (Pieczonka and Nuimura), I can only add a few general points that
the authors might wish to address:

1) A very brief outline of the Hexagon/Corona pipeline (Maurer and Rupper, 2015)
would be helpful.

We now include a brief summary of the process in section 2.1 on P4.

2) Why are only 21 glaciers studied? And what are the impacts of the 30% coverage
threshold? Previous geodetic studies (e.g. Gardelle et al., 2013, Bolch et al., 2011)
consider the entire glacierized area that is covered within a region. Using only 21
glaciers (and only those larger than 3 km2) and replacing potentially large missing areas
with the regional mean for a specific glacier type and elevation band could result in
biased regional estimates of glacier mass change that are not comparable to previous
studies.

In short, low radiometric contrast, cloud cover, and spatially correlated noise/error in the DEMs prevent
accurate calculation of changes for all glaciers in the region.  While this does limit direct comparison to
other previous studies (this among other things, such as different timespans covered), we feel these 21
large glaciers give a good regional picture of thickness changes over the 3 decade timespan.  We have
updated the discussion to more accurately reflect these facts in the paragraph starting on P6 line 14.

3) How is the ELA defined in this study (it first appears on P10L29)? Strictly speaking,
this is typically taken from surface mass balance measurements. While the elevation
that divides geodetic mass gain and loss would be related to the ELA, I am not sure that
it can be used as an ELA substitute (though I would be interested to hear otherwise).

We agree the term "ELA" was being used too loosely here, and instead substitute the term "glacier
hypsometry." Updated on P11 L20.

Specific comments:

P2L20: are these annual or seasonal streamflow contributions?

These are seasonal, their samples (from which their streamflow contributions were derived) were collected during September (post-monsoon). We have updated the text to make this important distinction clear (nice catch by the reviewer).

The text has been updated on P2 L19.

P4L20: Define DN.

DN = Digital Numbers, these are simply the pixel values in Landsat and ASTER images before being converted to reflectance or radiance. Updated on P4 L28.

P10L25: What data support the conclusion that debris-covered glaciers melt at the same rate as clean-ice glaciers? If this is overall mass balance rates than it should be specified. Figure 4 clearly shows that melt rates at debris-covered glaciers are lower than those observed on clean ice glaciers for the same elevation band, and this is later referenced by the authors on P10L28.

We now clarify at the beginning of the section, that although elevation distributions of ice loss differ between clean-ice and debris-covered glacier groups, overall geodetic mass balance values are similar in magnitude. Updated on P11 L16.

P11L15: Debris cover will almost always get thinner moving up-glacier. The greater thinning rates observed at the transition between debris-covered and debris-free zones is due in part to enhanced melt rates under thin debris cover but also due to the simple fact that bare ice will melt at a faster rate than debris-covered ice at the same elevation. Modelling studies in the Khumbu region (Shea et al., 2015; Rowan et al., 2015) both indicate that debris-covered tongues will detach from their accumulation areas in the future, leading to greater future melt rates.

We agree, and now include this statement and accompanying references in the text on P12 L12.

**Quantifying ice loss in the eastern Himalayas since 1974 using declassified spy satellite imagery**

Joshua M. Maurer[1,2], Summer B. Rupper[3], Joerg M. Schaefer[1,2]

[1]Lamont-Doherty Earth Observatory (L-DEO), Palisades, NY 10964, USA
[2]Department of Earth and Environmental Sciences, Columbia University, New York, New York 10027, USA
[3]University of Utah, Department of Geography, Salt Lake City, UT 84112, USA

*Correspondence to*: Joshua M. Maurer (jmaurer@ldeo.columbia.edu)

**Abstract.** Himalayan glaciers are important natural resources and climate indicators for densely populated regions in Asia. Remote sensing methods are vital for evaluating glacier response to changing climate over the vast and rugged Himalayan region; yet many platforms capable of glacier mass balance quantification are somewhat temporally limited considering typical glacier response times. We here rely on declassified spy satellite imagery and ASTER data to quantify surface lowering, ice volume change, and geodetic mass balance during 1974-2006 for glaciers in the eastern Himalayas, centered on the Bhutan-China border. The wide range of glacier types allows for the first mass balance comparison between clean, debris, and lake-terminating (calving) glaciers in the region. Measured glaciers show significant ice loss, with an estimated mean annual geodetic mass balance of -0.13 ± 0.06 m.w.e. yr$^{-1}$ (meters of water equivalent per year) for 10 clean-ice glaciers, -0.19 ± 0.11 m.w.e. yr$^{-1}$ for 5 debris-covered glaciers, -0.28 ± 0.10 m.w.e. yr$^{-1}$ for 6 calving glaciers, and -0.17 ± 0.05 m.w.e. yr$^{-1}$ for all glaciers combined. Contrasting hypsometries along with melt pond, ice cliff, and englacial conduit mechanisms result in statistically similar mass balance values for both clean-ice and debris-covered glacier groupsThe similar mass balances for clean-ice and debris-covered glaciers suggests that melt pond, ice cliff, and englacial conduit mechanisms are likely playing important roles in the melt process for debris-covered glaciers. Calving glaciers comprise 18% (66 km$^2$) of the glacierized area, yet have contributed 30% (-0.7 km$^3$) to the total ice volume loss, highlighting the growing relevance of proglacial lake formation and associated calving for the future ice mass budget of the Himalayas as the number and size of glacial lakes increase.

**1 Introduction**

Glaciers in high mountain Asia hold the largest store of ice outside the Polar Regions and contribute meltwater used by roughly 20 percent of the world's population for agriculture, energy production, and potable water (Immerzeel et al, 2010). Glacier changes must be quantified in order to evaluate impacts to hydrology and ecosystems, assess glacial lake outburst flood (GLOF) hazards, calculate recent contributions to sea level rise, and increase predictive capabilities regarding future change and resulting impacts.

**Commented [J1]:** Updated processing procedures (see section 2.3) now result in moderately different numbers here. Excluding steep non-glacial terrain in the accumulation zones (see page 5 line 25) had the largest effect on the mass balance value reported for debris-covered glaciers. However, these moderate changes do not affect the conclusions of the paper.

**Commented [J2]:** See comments on page 9 lines 10-17

[revised manuscript text omitted]
 without extrapolation, $\Delta V_{extrap}$ is ice volume change with after extrapolating missing data using regional data from individual elevation bandson. $\bar{h}$ is the spatially-averaged elevation change of the glacier (after extrapolation), and $\dot{b}$ is the geodetic mass balance for each glacier over the 32-year timespan (after extrapolation).

**Table S3**

| | clean | debris | calving | all |
|---|---|---|---|---|
| **Area (km²)** | 221 ± 11 | 78 ± 4 | 66 ± 3 | 365 ± 12 |
| **Area (%)** | 61 | 21 | 18 | |
| **Assuming zero change for missing data** | | | | |
| **ΔV (km³)** | -1.09 ± 0.4 | -0.55 ± 0.4 | -0.70 ± 0.3 | -2.34 ± 0.6 |
| **ΔV (%)** | 46 | 24 | 30 | |
| **b˙ (m.w.e. yr⁻¹)** | -0.13 ± 0.06 | -0.19 ± 0.11 | -0.28 ± 0.10 | -0.17 ± 0.05 |
| **Extrapolating missing data using regional profiles** | | | | |
| **ΔV (km³)** | -0.60 ± 0.4 | -0.62 ± 0.4 | -0.77 ± 0.3 | -1.99 ± 0.6 |
| **ΔV (%)** | 30 | 31 | 39 | |
| **b˙ (m.w.e. yr⁻¹)** | -0.07 ± 0.06 | -0.21 ± 0.11 | -0.31 ± 0.10 | -0.14 ± 0.05 |
| **Assuming zero change for missing data in glacier c, extrapolating missing data for all other glaciers** | | | | |
| **ΔV (km³)** | -0.95 ± 0.4 | -0.62 ± 0.4 | -0.77 ± 0.3 | -2.34 ± 0.6 |
| **ΔV (%)** | 41 | 26 | 33 | |
| **b˙ (m.w.e. yr⁻¹)** | -0.11 ± 0.06 | -0.21 ± 0.11 | -0.31 ± 0.10 | -0.17 ± 0.05 |

[Figure]

**Figure S1. Plots of elevation change vs. elevation, slope, maximum curvature, and ASTER along-track and cross-track directions for assumed stable terrain in each of the 8 Hexagon DEM regions given in Table S1. Black curves and grey shaded regions indicate the mean and standard deviation of each bin, respectively. The area (km$^2$) contained in each bin is indicated by the blue histogram bars, calculated as the number of pixels per bin * pixel resolution$^2$.**

[Figure]

**Figure S2.** Ice thickness change profiles for individual glaciers. Similar to Figure 4, Thickness changes are separated into 100 m bins, and the horizontal red lines indicate zero change.

[Figure]

**Figure S3. Hexagon and ASTER images, along with thickness change map processing stages for clean ice glaciers. Stage 1: raw elevation change maps; stage 2: after excluding erroneous pixels (see section 2.3), stage 3: after interpolating gaps smaller than 2 km², and stage 4: after filling remaining accumulation zone gaps with zero elevation change.**

[Figure]

Figure S4. Same as Fig. S3, but for debris-covered glaciers.

[Figure]

Figure S5. Same as Fig. S3, but for calving glaciers.

[Figure]

**Figure S6. Two examples of unstable moraine ridges. Red dotted ellipses indicate sections which have collapsed near glaciers a and k.**

[Figure]

**Figure S7.  Same as Fig. 3, except elevation changes are visualized as discrete classes rather than using continuous color coding.**

---

## Author Response (AR2)

Editor Decision: Publish subject to technical corrections (01 Aug 2016) by Dr. Etienne Berthier

Comments to the Author:

Dear Authors,

I have now read your revised manuscript and yours responses to the referees. The three referees were positive about your study and I think you addressed satisfyingly their comments.

I still have myself some minor, mostly technical, comments on your manuscript before it can be published. You will find them below. Thanks you for taking them into account carefully before uploading the final files for copy-editing.

Thanks a lot for choosing TC to publish your work.

Best regards,

Etienne Berthier

Editor's comments:

P2. L13. You may further add here that the glaciers surveyed in the field tend to be located in sub-regions where the mass loss is greater than in the region as a whole (as shown by Figure 5 And Figure S10 in (Gardner et al., 2013))

We now include this on page 2 line 14

P2L18. cap letter at the end of Vokso?

Fixed on page 2 line 19

P2L20. "during"

Fixed on page 2 line 21

P2. L31. "next few decades" a bit vague. Give the exact period over which a 10% area decrease is expected.

The statement "over the next few decades" was in error, as the 10% decrease in area is not given over a specified time interval, but rather as resulting from a climate forcing. Specifically, if climate were to remain at present day mean values. Updated on page 2 line 31

P2L34. space missing before "for"

Fixed on page 3 line 1

P5 L13. A max slope of 45° is indicated here but L6 of the same page it is stated that pixels located on slopes > 30° are excluded. Unclear how the two slope threshold relates to each others. Clarify.

On close review of our methods, we have realized that the 45° threshold is redundant, since we already exclude slopes > 30°. We have updated the text in the first paragraph of section 2.3 on page 5.

P5 L24. another impressive tools is the use of SAR coherence. Worth mentioning in this context (probably more powerful than InSAR velocity mapping): see for example (Frey et al., 2012)

We agree, and have updated the text on page 5 line 26.

P6 L15. remove "."

Fixed

P9 L13. Provide the time span over which the net mass balance is modeled in this study.

Now included on page 9 line 28

P9L20 "insulating"

Fixed on page 10 line 3

P11L12. add "elevation" between "covered" and "change"

Fixed on page 11 line 27 (used "thickness change" instead of "elevation change")

P11 L21. Could you back up (maybe downplay?) the statement that "debris-covered glaciers tend to have large accumulation areas" with a reference or based on your data? In the Khumbu area for example, the large debris-covered Khumbu and Ngzompa glaciers have also very large accumulation areas. So the statement does not appear to be universal. Maybe tell this is true for your selection of glaciers not everywhere.

We agree that the hypsometry observations for clean vs. debris-covered glaciers are likely not universal, and have updated the text on page 12 lines 2-10.

References for my comments

[revised manuscript text omitted]
$^2$) | 13.4 ± 1.3 | 25.0 ± 2.5 | 86.2 ± 8.6 | 23.5 ± 2.3 | 8.8 ± 0.9 | 30.1 ± 3.0 | 5.4 ± 0.5 |
| ΔV (km$^3$) | -0.08 ± 0.03 | -0.29 ± 0.07 | -0.26 ± 0.31 | -0.14 ± 0.06 | -0.09 ± 0.03 | -0.17 ± 0.09 | -0.03 ± 0.03 |
| ΔV$_{extrap}$ (km$^3$) | -0.05 ± 0.03 | -0.27 ± 0.07 | 0.09 ± 0.31 | -0.20 ± 0.06 | -0.09 ± 0.03 | -0.18 ± 0.09 | -0.03 ± 0.03 |
| $\bar{h}$ (m) | -5.6 ± 2.2 | -12.0 ± 3.2 | -3.0 ± 3.6 | -6.1 ± 2.5 | -10.3 ± 4.0 | -5.5 ± 2.9 | -5.2 ± 5.9 |
| $\dot{b}$ (m.w.e.) | -0.15 ± 0.06 | -0.32 ± 0.09 | -0.08 ± 0.10 | -0.16 ± 0.07 | -0.27 ± 0.11 | -0.15 ± 0.08 | -0.14 ± 0.16 |
| Data coverage (%) | 39 | 48 | 28 | 49 | 100 | 68 | 100 |
| Debris coverage (%) | 1 | 6 | 3 | 41 | 37 | 44 | 16 |
| Calving (y/n) | n | y | n | n | n | n | y |

| | h | i | j | k | l | m | n |
|---|---|---|---|---|---|---|---|
| | 90.27 | 90.33 | 90.35 | 90.39 | 90.47 | 90.7 | 90.75 |
| | 28.13 | 28.11 | 28.09 | 28.1 | 28.08 | 28.06 | 28.04 |
| | 5486 | 5151 | 5154 | 5749 | 6133 | 5505 | 5450 |
| | 13.8 ± 1.4 | 6.1 ± 0.6 | 5.0 ± 0.5 | 49.8 ± 5.0 | 29.7 ± 3.0 | 5.7 ± 0.6 | 9.2 ± 0.9 |
| | -0.12 ± 0.08 | -0.10 ± 0.03 | -0.04 ± 0.02 | -0.52 ± 0.13 | -0.02 ± 0.13 | -0.02 ± 0.02 | -0.08 ± 0.03 |
| | -0.16 ± 0.08 | -0.10 ± 0.03 | -0.08 ± 0.02 | -0.45 ± 0.13 | 0.07 ± 0.13 | -0.02 ± 0.02 | -0.09 ± 0.03 |
| | -8.5 ± 5.7 | -18.3 ± 6.0 | -8.7 ± 4.8 | -10.7 ± 2.8 | -0.6 ± 4.3 | -4.4 ± 3.3 | -8.4 ± 3.1 |
| | -0.23 ± 0.15 | -0.48 ± 0.16 | -0.23 ± 0.13 | -0.28 ± 0.08 | -0.02 ± 0.11 | -0.12 ± 0.09 | -0.22 ± 0.08 |
| | 30 | 100 | 58 | 54 | 26 | 60 | 76 |
| | 17 | 11 | 10 | 11 | 14 | 1 | 1 |
| | y | y | n | n | n | n | y |

| | o | p | q | r | s | t | u |
|---|---|---|---|---|---|---|---|
| | 90.79 | 90.78 | 90.62 | 90.66 | 90.68 | 90.63 | 90.67 |
| | 28.03 | 28.06 | 28.21 | 28.24 | 28.26 | 28.25 | 28.29 |
| | 5216 | 5540 | 5342 | 5139 | 5304 | 6034 | 5602 |
| | 5.7 ± 0.6 | 9.1 ± 0.9 | 3.1 ± 0.3 | 12.5 ± 1.3 | 6.5 ± 0.7 | 9.8 ± 1.0 | 10.2 ± 1.0 |
| | -0.04 ± 0.02 | -0.09 ± 0.02 | -0.04 ± 0.02 | -0.11 ± 0.04 | -0.03 ± 0.03 | -0.03 ± 0.03 | -0.04 ± 0.03 |
| | -0.07 ± 0.02 | -0.12 ± 0.02 | -0.04 ± 0.02 | -0.11 ± 0.04 | -0.03 ± 0.03 | -0.01 ± 0.03 | -0.05 ± 0.03 |
| | -7.6 ± 3.4 | -10.1 ± 3.0 | -11.9 ± 6.4 | -9.1 ± 3.1 | -5.1 ± 4.4 | -3.2 ± 3.4 | -4.3 ± 2.9 |
| | -0.20 ± 0.09 | -0.27 ± 0.08 | -0.32 ± 0.17 | -0.24 ± 0.09 | -0.13 ± 0.12 | -0.08 ± 0.09 | -0.11 ± 0.08 |
| | 63 | 41 | 100 | 84 | 100 | 55 | 51 |
| | 8 | 1 | 42 | 39 | 1 | 19 | 9 |
| | n | y | n | n | n | n | n |

$\Delta V$ is ice volume change without extrapolation, $\Delta V_{extrap}$ is ice volume change after extrapolating missing data using regional data from individual elevation bands. $\bar{h}$ is the spatially-averaged elevation change of the glacier, and $\dot{b}$ is the geodetic mass balance for each glacier over the 32-year timespan.

| | clean | debris | calving | all |
|---|---|---|---|---|
| **Area (km²)** | 221 ± 11 | 78 ± 4 | 66 ± 3 | 365 ± 12 |
| **Area (%)** | 61 | 21 | 18 | |
| **Assuming zero change for missing data** | | | | |
| **ΔV (km³)** | -1.09 ± 0.4 | -0.55 ± 0.4 | -0.70 ± 0.3 | -2.34 ± 0.6 |
| **ΔV (%)** | 46 | 24 | 30 | |
| **b˙ (m.w.e. yr⁻¹)** | -0.13 ± 0.06 | -0.19 ± 0.11 | -0.28 ± 0.10 | -0.17 ± 0.05 |
| **Extrapolating missing data using regional profiles** | | | | |
| **ΔV (km³)** | -0.60 ± 0.4 | -0.62 ± 0.4 | -0.77 ± 0.3 | -1.99 ± 0.6 |
| **ΔV (%)** | 30 | 31 | 39 | |
| **b˙ (m.w.e. yr⁻¹)** | -0.07 ± 0.06 | -0.21 ± 0.11 | -0.31 ± 0.10 | -0.14 ± 0.05 |
| **Assuming zero change for missing data in glacier c, extrapolating missing data for all other glaciers** | | | | |
| **ΔV (km³)** | -0.95 ± 0.4 | -0.62 ± 0.4 | -0.77 ± 0.3 | -2.34 ± 0.6 |
| **ΔV (%)** | 41 | 26 | 33 | |
| **b˙ (m.w.e. yr⁻¹)** | -0.11 ± 0.06 | -0.21 ± 0.11 | -0.31 ± 0.10 | -0.17 ± 0.05 |

[Figure]

**Figure S1.** Plots of elevation change vs. elevation, slope, maximum curvature, and ASTER along-track and cross-track directions for assumed stable terrain in each of the 8 Hexagon DEM regions given in Table S1. Black curves and grey shaded regions indicate the mean and standard deviation of each bin, respectively. The area (km$^2$) contained in each bin is indicated by the blue histogram bars, calculated as the number of pixels per bin * pixel resolution$^2$.

[Figure]

[Figure]

**Figure S2. Ice thickness change profiles for individual glaciers. Similar to** Figure 4, **Thickness changes are separated into 100 m bins, and the horizontal red lines indicate zero change.**

[Figure]

**Figure S3. Hexagon and ASTER images, along with thickness change map processing stages for clean ice glaciers. Stage 1: raw elevation change maps; stage 2: after excluding erroneous pixels (see section 2.3), stage 3: after interpolating gaps smaller than 2 km², and stage 4: after filling remaining accumulation zone gaps with zero elevation change.**

[Figure]

**Figure S4. Same as Fig. S3, but for debris-covered glaciers.**

[Figure]

**Figure S5. Same as Fig. S3, but for calving glaciers.**

[Figure]

**Figure S6. Two examples of unstable moraine ridges. Red dotted ellipses indicate sections which have collapsed near glaciers a and k.**

[Figure]

[Figure]

[Figure]

meters

**Figure S7.  Same as Fig. 3, except elevation changes are visualized as discrete classes rather than using continuous color coding.**